# Increased ectodysplasin-A2-receptor EDA2R is a ubiquitous hallmark of aging and mediates parainflammatory responses

Maria Chiara Barbera [1,2,20], Luca Guarrera [1,20], Andrea David Re Cecconi [3,20], Giada Andrea Cassanmagnago [1,4,5], Arianna Vallerga [1], Martina Lunardi[3], Francesca Checchi[1,6], Laura Di Rito[1], Margherita Romeo [7], Sarah Natalia Mapelli [8], Benedikt Schoser[9], Edward V. Generozov[10], Molecular Genetics Group*, Rick Jansen [11,12,13], Eco J. C. de Geus [14], Brenda Penninx [11,12,13], Jenny van Dongen[14], Ilaria Craparotta[1], Rosanna Piccirillo [3], Ildus I. Ahmetov [15,16,17,21] & Marco Bolis [1,4,5,18,21] ✉

Intensive efforts have been made to identify features that could serve as biomarkers of aging. Yet, drug-based interventions aimed at lessening the detrimental effects of getting older are lacking. This is largely attributable to tissue-specificity, sex-related differences, and to the difficulty of identifying actionable targets, which continues to pose a significant challenge. Here, we implement a bioinformatics approach revealing that aging-associated increase of the transmembrane Ectodysplasin-A2-Receptor is a prominent tissue-independent alteration occurring in humans and other species, and is particularly pronounced in models of accelerated aging. We show that strengthening of the Ectodysplasin-A2-Receptor signalling axis in myogenic precursors and differentiated myotubes suffices to trigger potent parainflammatory responses, mirroring aspects of aging-driven sarcopenia. Intriguingly, obesity, insulin-resistance, and aging-related comorbidities, such as type-2-diabetes, result in heightened levels of the Ectodysplasin-A2 ligand. Our findings suggest that targeting the Ectodysplasin-A2 surface receptor represents a promising pharmacological strategy to mitigate the development of aging-associated phenotypes.

Ectodysplasin A2 Receptor (EDA2R) is a member of the tumor necrosis factor receptor (TNFR) superfamily which selectively binds to EDA-A2, a protein encoded by an alternative splicing isoform of *EDA* (Ectodysplasin A) gene. EDA2R receptor has been recognized as a target of TP53[1], and EDA2R/EDA-A2 signaling has been observed to mediate activation of JNK, NF-kB pathways[2,3] and to promote apoptosis and cell death[4]. Moreover, *EDA2R* mRNA expression was reported to be elevated in the aging lungs[5], and several studies indicated that

polymorphisms in the *EDA2R* gene locus are linked with age-associated androgenetic alopecia (AGA)[6].

Despite these observations, the broader role of EDA2R in aging remains poorly understood. In this work, we identify EDA2R as a key tissue-independent marker of aging through the analysis of multi-tissue transcriptomics datasets. We further demonstrate its functional relevance in skeletal muscle, where increased EDA2R expression induces parainflammatory responses, mimicking aspects of aging-

A full list of affiliations appears at the end of the paper. *A list of authors and their affiliations appears at the end of the paper. ✉e-mail: marco.bolis@ior.usi.ch; marco.bolis@marionegri.it

driven sarcopenia. These findings highlight the EDA2R/EDA-A2 axis as a promising pharmacological target for mitigating aging-associated phenotypes.

## Results and discussion

### EDA2R correlates with age across tissues and species

To identify genes whose expression positively correlates with the increasing age of individuals, we selectively screened the Genotype-Tissue Expression database (GTEX)[7] for transcriptional changes occurring in a tissue-independent fashion. For this purpose, we employed a rigorous multi-step bioinformatics analysis procedure which aimed at minimizing possible confounding factors, such as gender differences and the precise anatomical sub-localization of the sample of origin (Methods). This analysis incorporated a Leave-Half-Out (LHO) resampling approach and outlier removal to ensure the robustness of correlation results across tissues (Supplementary Fig. 1). *EDA2R* emerged as the most noticeable outlier of the analysis, as its expression resulted to be strongly correlated to the increasing age of donors in all organs. Indeed, *EDA2R* appeared among the very top hits in all solid tissues evaluated both in males and females (Fig. 1a, Supplementary Figs. 2–5, Supplementary Data 1-2), and its mean correlation coefficient ranked first in the integrated pan-tissue analysis (Fig. 1b). Conversely, no widespread tissue-agnostic trends were observed for the gene encoding for the other ectodysplasin-A receptor (EDAR) nor for other members of the tumor necrosis factor superfamily (Supplementary Fig. 6a). A strong increase of *Eda2r* expression over the course of lifespan also emerged in tissues of rats (*Rattus norvegicus*) and mice (*Mus musculus*), suggesting a non-species-specific behavior (Fig. 1c, Supplementary Data 3-4).

### EDA2R is elevated in models of accelerated aging

To further strengthen the connection with age, we investigated transcriptional changes occurring in murine models of Hutchinson-Gilford progeria syndrome (HGPS), a recognized proxy of accelerated aging in humans[8–10]. In this context, *Eda2r* emerged as one of the most significantly up-regulated genes in HGPS mice compared to age-matched wild-type counterparts (Fig. 1d, e, Supplementary Data 5), thereby emphasizing the link with the aging process across different tissues and cell-types. The marked up-regulation of *EDA2R* in the aortic artery of HGPS mice (Fig. 1d) is particularly relevant, as the circulatory system is crucial in cardiovascular health. Aging and age-related diseases often cause vascular dysfunction, and increased expression of *EDA2R* may reveal a mechanism by which aging and progeroid conditions contribute to vascular pathology. Collectively, our observations strongly support a unique tissue-independent association between aging and *EDA2R* mRNA levels, prompting further investigation into whether increased *EDA2R* expression could contribute to the development of aging phenotypes.

### EDA2R mediates inflammatory signaling in skeletal muscle

Recent research has unveiled that elevated *EDA2R* expression in skeletal muscle is crucial in promoting cancer-induced muscle-loss (cachexia)[11,12], which mirrors several aspects of aging-associated muscle atrophy known as sarcopenia[13]. Given the pronounced correlation between *EDA2R* mRNA levels and age in skeletal muscle of humans (Fig. 2a, Supplementary Fig. 6b, Supplementary Data 6), mice (Fig. 2b, Supplementary Data 7) and rats (Fig. 2c, Supplementary Data 8), we ought to prove whether heightened *EDA2R* expression could contribute to drive the transcriptional alterations observed with aging. To guide the selection of the most appropriate cellular model for functional experiments, we analyzed single-cell RNA-Seq data from various studies, encompassing a large number of cells from the skeletal muscle of mice[14,15] (Fig. 2d) and humans[16,17] (Fig. 2e, Supplementary Fig. 6c). In this context, we observed a positive correlation between *EDA2R* expression and age across various cell populations, confirming that the

aging-associated increase of this receptor is cell-type independent (Fig. 2e). Notably, we found that transcription of *EDA2R* is particularly pronounced in myotubes and myogenic precursors (satellite cells/myoblasts), which play a crucial role in muscle development and repair. Indeed, alterations in function and number of these progenitors can contribute to the age-related decline in muscle mass and strength[18]. Therefore, we overexpressed *Eda2r* in undifferentiated murine myoblasts (Supplementary Fig. 6d–f) and observed that the sole increase of *Eda2r* is sufficient to induce potent transcription of chemoattractants (i.e. *Cxcl5*) and pro-inflammatory factors (i.e., *Il6*), mirroring transcriptional alterations seen in aging muscle (Fig. 3a, Supplementary Data 9). In addition, overexpression of *EDA2R* in terminally differentiated myotubes (Supplementary Fig. 6g–i, Supplementary Data 10) resulted in perturbations resembling those observed in myogenic precursors (Fig. 3a–c), including repression of pathways associated with muscle development, increased autophagy and protein catabolism (Supplementary Fig. 6i). Importantly, the overexpression of *EDA2R* in human myoblasts (Supplementary Fig. 6k, Supplementary Data 11) recapitulated the results obtained in mice (Fig. 3a, c, d), indicating that downstream transcriptional responses are consistent across species. Conversely, inhibition of *EDA2R* in the same model significantly reduced the transcription of pro-inflammatory factors (Supplementary Fig. 6k), highlighting the receptor's critical role in mediating immune responses and suggesting a potential therapeutic target for mitigating inflammation-related conditions.

### EDA2R/EDA-A2 links aging to comorbidity-driven inflammation

Building on these findings, we extended our research to investigate whether similar age-related transcriptional changes in *EDA2R* are observed in blood, utilizing large-scale transcriptomics data from two extensive cohorts: the Netherlands Study of Depression and Anxiety (NESDA, *n* = 2064) and the Netherlands Twin Register (NTR, *n* = 3164). Our analysis confirmed the correlation with age and revealed a positive association between *EDA2R* expression in blood and the abundance of circulating C-reactive protein, a well-known marker of systemic inflammation (Fig. 3e). Moreover, the age-related increase in *EDA2R* is confirmed at the protein level in plasma-derived samples[19], indicating that the observed effects are not solely transcriptional[20] (Supplementary Fig. 7). Hence, considering the age-related, tissue-agnostic rise in *EDA2R* levels and its potential to trigger pro-inflammatory responses, its activation and its downstream signaling become particularly noteworthy. Ignition of EDA2R/EDA-A2 signaling occurs due to either increased EDA2R or elevated EDA-A2 levels, both leading to the activation of non-canonical NF-kB signaling[3,21] (Supplementary Fig. 6l). Indeed, transcriptional perturbations induced by EDA-A2 treatment closely mimic those resulting from overexpression of *EDA2R* (Fig. 3h, Supplementary Data 12), suggesting that similar effects can be achieved by targeting the signaling pathway from both sides. Notably, we found that RNA levels of *EDA-A2* progressively rise with age in blood-derived samples (Fig. 3f, g), with T-Cells predicted as the main source (Supplementary Data 13). Hence, we examined whether this transcriptional increase is reflected at the protein level, considering that mass spectrometry cannot differentiated between EDA-A1 and EDA-A2 due to their high sequence similarity. While proteomic analysis showed that total EDA levels in plasma remain stable with age (Supplementary Fig. 8), they significantly rise in conditions like type 2 diabetes[22,23], fatty liver disease[24,25], insulin resistance and obesity[26], which have all been linked to the increased risk of developing sarcopenic phenotypes[27–32]. Intriguingly, sarcopenia itself was associated with heightened levels of EDA-A2 in mice[33]. These findings suggest that although blood-produced EDA may represent only a small fraction of total circulating EDA, various aging-related comorbidities could further amplify EDA2R/EDA-A2 pathway activation by influencing

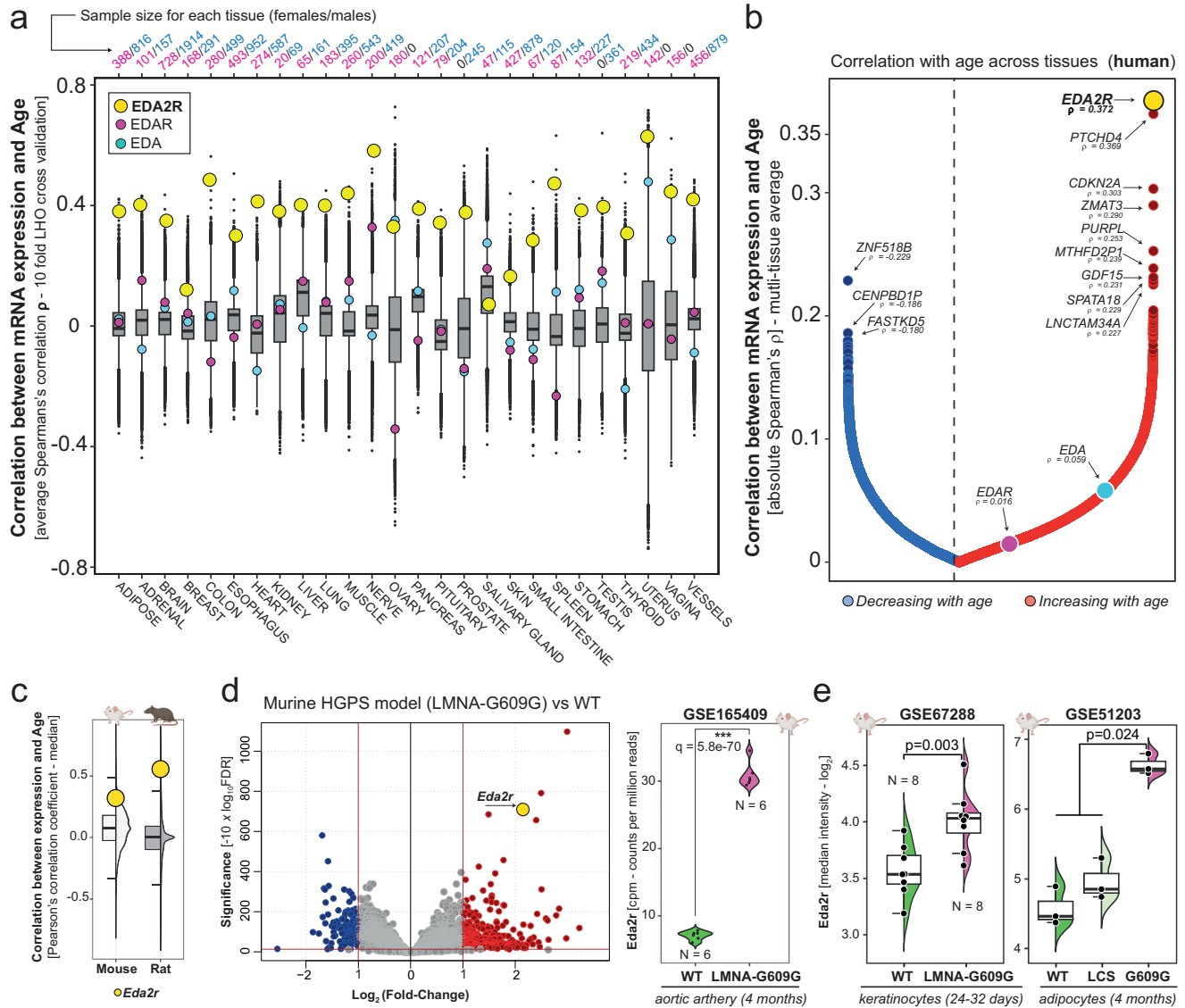

**Fig. 1 | Cross-species correlation of EDA2R expression with age and upregulation in premature aging models. a** Boxplots representing average Spearman's correlation coefficients between mRNA expression of all genes with age, from the GTEX dataset. Correlations were calculated across 10-fold Leave-Half-Out random resampling per tissue. Relative rankings of *EDA2R*, *EDAR*, and *EDA* are indicated. Sample sizes for each tissue stratified by sex are shown at the top. **b** Representation of the mean Spearman's correlation coefficients (ρ) for each gene across solid tissues, with red and blue dots indicating genes that increase or decrease with age, respectively. **c** Boxplots representing the median Pearson's correlation coefficients between age and mRNA expression across tissues of mice (lightgrey) and rats (darkgrey), as determined from Tabula Muris Senis (GSE132040 dataset, 14 tissues, *n* = 850 distinct samples) and Rat BodyMap (GSE53960 dataset, 11 tissues, *n* = 320 distinct samples). Mouse and rat icons: Created in BioRender. Bolis, M. (2025) https://BioRender.com/h42n006. **d** Left, Volcano-plot representing differential expression in murine aortic artery of HGPS, 2 months old mice (LMNA-G609G mutant, *n* = 6) versus wild-type, 2 months old age-matched mice (GSE165409,

*n* = 6). Blue and red colors indicate down or upregulated genes with a log₂ fold-change lower than -1 or greater than 1 and an FDR < 0.05. Right, Boxplots representing *Eda2r* expression in the aortic artery of HGPS (pink, *n* = 6) and wild-type (green, *n* = 6) mice. *P*-values were computed using Wald statistics (two-sided) as implemented in DESeq2, with *p*-values adjusted for multiple comparisons using the Benjamini-Hochberg procedure. Indicated is the exact FDR of *Eda2r q* = 5.8e−70. Mouse icon: Created in BioRender. Bolis, M. (2025) https://BioRender.com/i32c095. **e** Left, boxplots comparing microarray expression levels of *Eda2r* between HPGS (pink) and age-matched wild-type (green) mice in murine keratinocytes (GSE67288, *n* = 8 biological replicates in each group) and (right) gonadal adipocytes (GSE51203, *n* = 3 biological replicates in each group). Mutant mice deficient of Lamin C but not Lamin A (LCS, lightgreen) are provided as comparison. *P*-values were calculated using the Wilcoxon Rank-Sum test (two-sided) and adjusted for multiple comparisons using FDR. Mouse icon: Created in BioRender. Bolis, M. (2025) https://BioRender.com/m99g202. – Boxplots boundaries and source data are provided as a Source Data file.

ligand levels, thereby enhancing our understanding of EDA2R/EDA-A2 signaling in the context of aging (Fig. 3i).

Taken together, our results disclose that transcription of *EDA2R* emerges as a prominent feature associated with increasing age in solid tissues, with functional relevance in skeletal muscle. Currently, there is no established antagonist specifically targeting EDA2R, which limits direct pharmacological interventions. Overall, the EDA2R/EDA-A2 axis may represent a novel target for the development of compounds

which could provide new therapeutic avenues for managing inflammation and related conditions associated with aging phenotypes and the subsequent age-associated decline (Fig. 4).

## Methods

### Inclusion & ethics

This research was conducted with a commitment to inclusion and ethical practices. The research team included members from

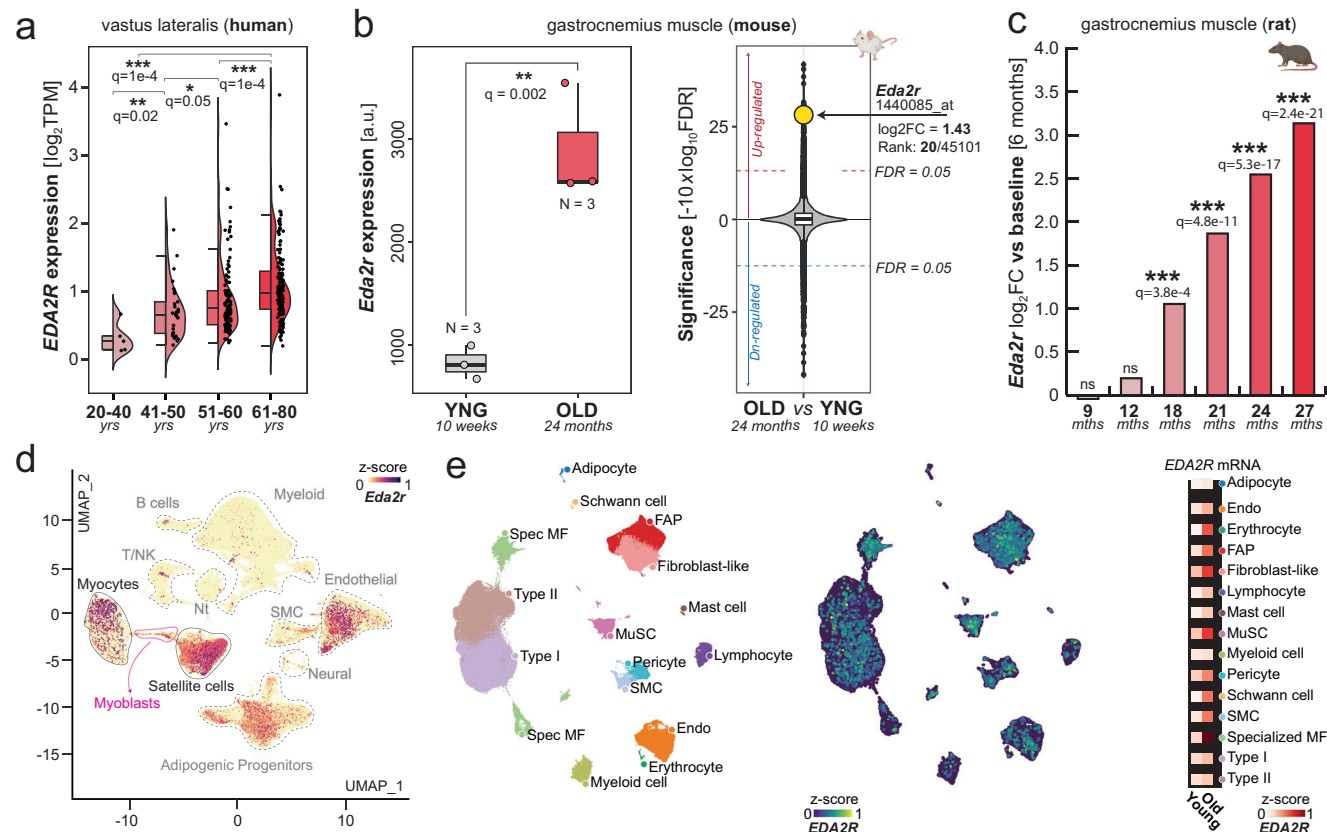

**Fig. 2 | *EDA2R Expression in Aging Muscle at Bulk and Single-Cell Levels.* a** mRNA expression (TPM) of *EDA2R* in human samples derived from vastus lateralis (*n* = 291, phs001048). Patients were categorized into 4 groups based on their age (20–40 *n* = 5; 41–50 *n* = 29; 51–60 *n* = 110; 61–80 *n* = 147). P-values were determined using Wilcoxon rank sum test (two-sided) and adjusted for multiple comparison using FDR. Exact q-values are provided. (*** *q* ≤ 001; ** *q* ≤ 0.01; * *q* ≤ 0.05). Darker shades of red represent older age groups. **b** Left, RMA-normalized expression of *Eda2r* measured by microarrays is indicated as normalized fluorescence intensity (arbitrary units: a.u.). Boxplots compare murine gastrocnemius samples derived from young (lightgrey) and old (red) mice (*n* = 3 biolgical replicates in each group, GSE52550). *P*-values were calculated using the Wilcoxon Rank-Sum test (two-sided). Statistical significance was adjusted using FDR. (*** *q* ≤ 001; ** *q* ≤ 0.01; * *q* ≤ 0.05; Exact *q*-value for *Eda2r* = 0.002). Right, Microarray probes ranked by their statistical significance derived from the same comparison. Mouse icon: Created in BioRender. Bolis, M. (2025) https://BioRender.com/i39v752. **c** Changes in *Eda2r*

expression in rat gastrocnemius muscle with age (GSE53960). Indicated *P*-values were computed using Wald statistics (two-sided) as implemented in DESeq2, with *p*-values adjusted for multiple comparisons using the FDR. (*** indicates FDR < 0.001). Darker shades of red represent older age groups. (6 months, *n* = 7; 9 months, *n* = 7; 12 months, *n* = 8; 18 months, *n* = 7; 21 months, *n* = 7; 24 months, *n* = 8; 27 months, *n* = 8). Rat icon: Created in BioRender. Bolis, M. (2025) https://BioRender.com/g32r397. **d** UMAP depicting *Eda2r* expression at single-cell resolution, including over 365,000 cells, in murine muscle as quantified from McKellar et al.[14](Nt Neutrophils, SMC smooth muscle cells). **e** UMAP showing single cells from human limb skeletal muscles, as quantified by Lai et al.[17], including over 387,000 cells/nuclei from individuals aged 15 to 99 years. Left: Major cell types annotated (MF Myofiber, Endo Endothelial, MuSC muscle stem cells, SMC smooth muscle cells, FAP fibro-adipogenic progenitors). Center: Expression of *EDA2R*. Right: *EDA2R* expression across cell-types in young and old individuals. – Boxplots boundaries and source data are provided as a Source Data file.

diverse backgrounds, with a focus on promoting equity in scientific inquiry. Our research was designed to be accessible and beneficial to the global scientific community, with data and findings shared openly to promote broader understanding and application.

### Cell culture
C2C12 (ATCC, #CRL-1772), a myoblast cell line from the C3H mouse strain, was grown in DMEM (Dulbecco's Modified Eagle's Medium, Gibco), supplemented with 10% fetal bovine serum (FBS) (Euroclone, Pero, Italy) and 2 mM L-glutamine (Gibco), and maintained in culture at 37 °C with 5% $CO_2$, without reaching confluence to avoid fusion into myotubes. Myoblasts differentiated into myotubes when they reached 80% confluence and were cultured for four days in DMEM, 2 mM L-glutamine (Gibco), supplemented with 2% horse serum (HS) (Euroclone, Pero, Italy), at 37 °C and 8% $CO_2$. The differentiation medium was changed every two days. Skeletal muscle myoblasts were isolated from a human male donor. Written

informed consent was obtained from the donor, and the study protocol was approved by the Ethical Review Committee at the Ludwig-Maximilians-University, Munich, Germany (IRB-No. 45-14). Cells were grown in Skeletal Muscle Cell Growth Medium with Supplement Mix (PromoCell, Heidelberg, Germany), and maintained in culture at 37 °C with 5% $CO_2$, without reaching confluence to avoid fusion into myotubes.

### Transfection of murine myoblasts by lipofectamine
C2C12 myoblasts were grown in a 6-well plate and transfected with pCMX-GFP plasmid (gifted from Prof. Kakizuka, Japan) or EDA2R-pCMV6-AC-GFP-expressing plasmid (MG222258, Origene) using Lipofectamine 2000 (Life Technologies), according to the manufacturer's directions. After 48 h, total RNA was isolated from cells with QIAzol Lysis Reagent (Qiagen) and extracted using miRNeasy Kit (Qiagen). RNA concentration was evaluated through Qubit RNA High Sensitivity Assay Kit (Invitrogen, Waltham, MA, USA) while RNA quality was established using 4200 Tapestation (Agilent Technologies, Santa Clara, CA, USA).

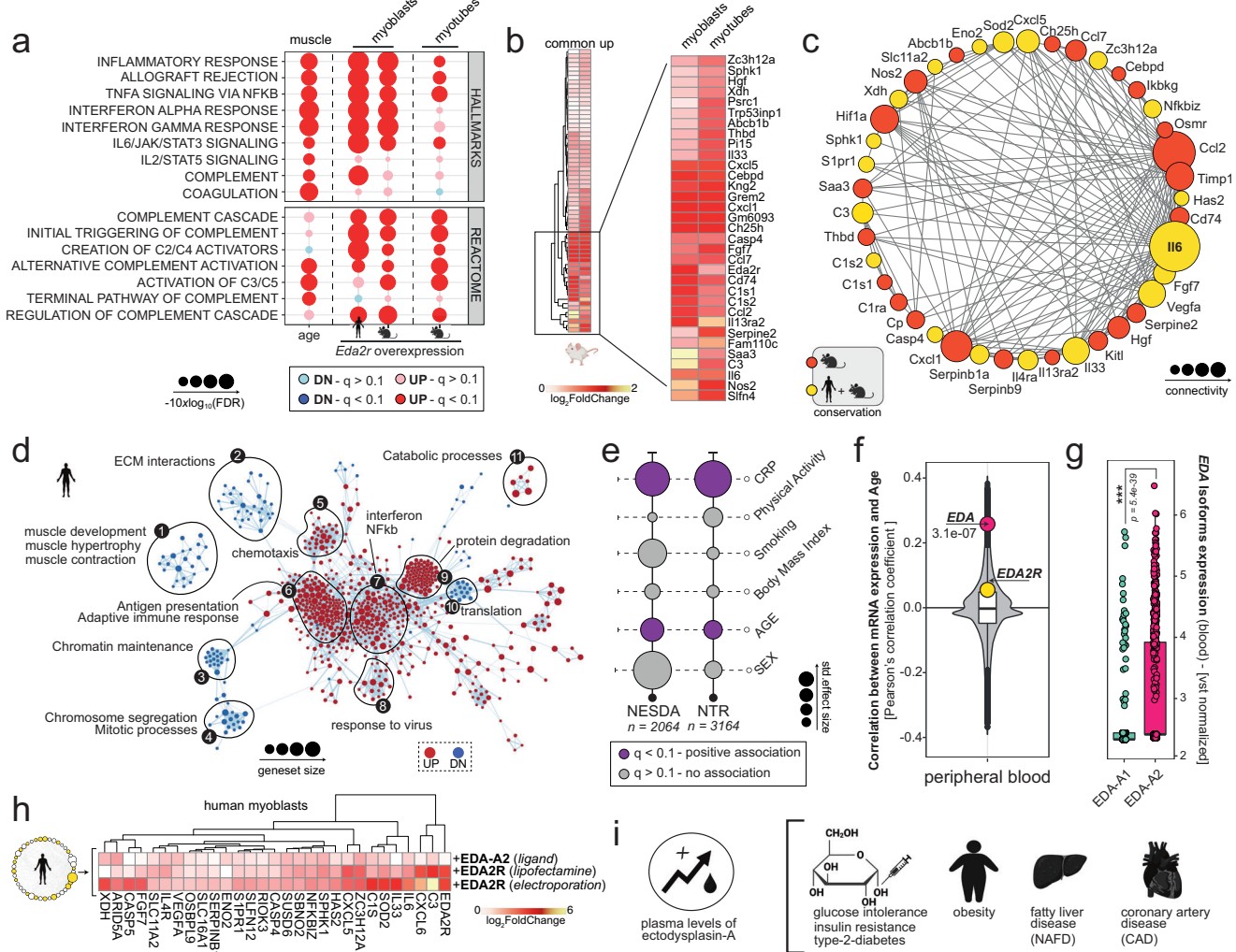

**Fig. 3 | Activation of EDA2R/EDA-A2 triggers parainflammatory responses.**
**a** Age-dependent (column 1), *EDA2R/Eda2r* overexpression-dependent (columns 2–4) enrichments in Hallmark and Reactome inflammation-related gene sets, as computed in aging human gastrocnemius muscle (GTEX dataset, column1) in human myoblasts (column 2), C2C12 murine myoblasts (column 3), and C2C12 differentiated murine myotubes (column 4). *P*-Values were adjusted for multiple testing using FDR. Human and mouse icons: Created in BioRender. Bolis, M. (2025) https://BioRender.com/s08q603. **b** Heatmap representing genes significantly upregulated (FDR < 0.05) in both murine myoblasts and differentiated murine myotubes following *Eda2r* overexpression. Highlighted genes exhibit higher levels of fold induction. Mouse icon: Created in BioRender. Bolis, M. (2025) https://BioRender.com/v79d694. **c** Resulting protein-protein interaction network (String-DB). Genes whose induction is confirmed in human myoblasts following *EDA2R* overexpression are indicated in yellow. Human and mouse icons: Created in BioRender. Bolis, M. (2025) https://BioRender.com/q91f320. **d** Enrichment map depicting significant induction or repression of gene sets from the comparison of transcriptional profiles of *EDA2R*-overexpressing vs. wild-type human myoblasts. Each dot represents an individual gene set, with the size of dots proportional to the size of each gene set. Edges between nodes represent shared genes between adjacent pathways. The analysis includes gene sets from the Hallmark, Reactome,

Wikipathways, and GeneOnotology (BP-biological processes) collections. Human icon: Created in BioRender. Bolis, M. (2025) https://BioRender.com/u50k325. **e** Dotplot representing associations between *EDA2R* expression in peripheral blood with Age, Sex, BMI, Smoking, Physical Activity and plasmatic levels of C-reactive protein. NESDA=Netherlands Study of Depression and Anxiety (*n* = 2064); NTR Netherlands Twin Registry (*n* = 3164). Statistical significance (fdr-adjusted q-value < 0.1) is indicated in violet, and dot dimension reflects the magnitude of the estimated effect size. **f** Violin-plot representing Pearson's correlations between expression and age for all human genes in venous blood (GTEX dataset, *n* = 497 distinct individuals). **g** Expression of *EDA-A1* (green) and *EDA-A2* (pink) isoforms (TPM) in human venous blood. Statistical significance was assessed using two-sided Wilcoxon non-parametric test (GTEX dataset, *n* = 755 distinct individuals). **h** Heatmap depicting the relative fold-change compared to matched controls of conserved genes identified in **c** across human myoblasts subjected to EDA-A2 ligand treatment, *EDA2R* overexpression by transfection, and *EDA2R* over-expression by electroporation. Human icon: Created in BioRender. Bolis, M. (2025) https://BioRender.com/j36h607. **i** Health conditions and aging-associated comorbidities linked to increased levels of plasmatic EDA. Created in BioRender. Bolis, M. (2025) https://BioRender.com/t14o510. – Boxplots boundaries and source data are provided as a Source Data file.

For experiments involving terminal differentiation, myoblast where allowed to reach confluence and cultured as previously described.

## Transfection of human myoblasts by lipofectamine or electroporation

Skeletal muscle myoblasts isolated from a human male donor were transfected using the Lipofectamine™ 2000 Transfection Reagent

(Invitrogen, 11668019). 24 h before transfections, cells were plated to a density of 17.500 cell/cm² in a 6-well plate. The transfection was performed according to the manufacturer's protocol. For each well, 4 µg of plasmid DNA (EDA2R, Origene) and the appropriate amount of Lipofectamine™ 2000 were diluted in Opti-MEM™ I Reduced Serum Medium (Gibco, 31985070). After transfection, cells were incubated at 37 °C in a humidified atmosphere containing 5% CO₂ for 24 h. For

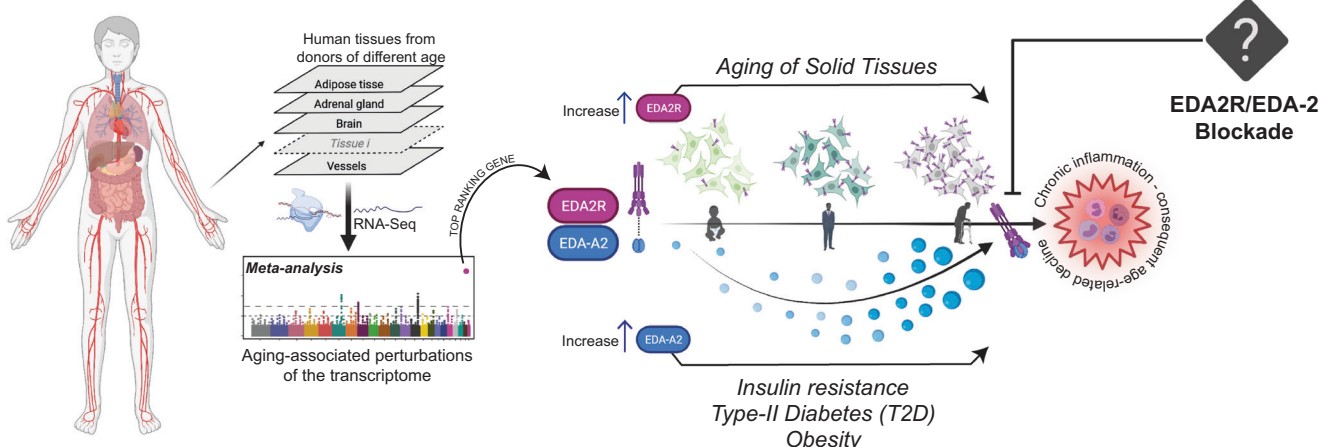

**Fig. 4 | Increase of EDA2R is reinforced by elevation of EDA-A2 in aging-associated comorbidities.** The figure summarizes our findings, illustrating the tissue-wide correlation analysis that identified *EDA2R* as the top-ranking gene associated with aging across multiple tissues. The observed increase in *EDA2R* expression is accompanied by elevated levels of its ligand, EDA-A2, under specific aging-related conditions such as insulin resistance, type 2 diabetes (T2D), and obesity. Enhanced activation of the EDA2R/EDA-A2 signaling axis triggers parainflammatory responses, representing a potential therapeutic target for intervention using EDA2R-specific antagonists. Created in BioRender. Bolis, M. (2025) https://BioRender.com/u99v533.

electroporation, skeletal muscle myoblasts isolated from a human male donor were transfected using the Amaxa™ Nucleofector Basic Kit for Primary Mammalian Smooth Muscle Cells (Lonza, VPI-1004). For each transfection condition, a total of 1 million cells were resuspended in the Nucleofector Solution provided with the kit. Then, 3 µg of plasmid DNA (EDA2R, Origene) was added to the cell suspension and subjected to nucleofection using the Amaxa Nucleofector Device with the A-033 program, as recommended by the manufacturer's protocol. Following nucleofection, the cells were immediately transferred into pre-warmed complete growth medium and distributed into 3 wells of 9.6 cm² each. The cells were then incubated at 37 °C in a humidified atmosphere containing 5% $CO_2$ for 48 h, until RNA extraction.

### Treatment of human myoblasts
Skeletal muscle myoblasts isolated from a human male donor were treated with EDA-A2 ligand (922-ED, R&D Systems) at the concentration of 250 ng/ml used in Bilgic et al.[11], and incubated for an additional 48 h, until RNA extraction.

### Libraries preparation
Following the TruSeq Stranded Total RNA protocol (Illumina, San Diego, CA, USA) at least 500 ng of RNA whose RIN value was between 7 and 9, were used for RNA sequencing. RNA sequencing was run on a NextSeq 500 sequencer (Illumina) using a 1 × 75 high-output flow cell with all 18 samples/run.

### Analysis workflow for human samples
RNA-Seq raw counts data were retrieved from GTEX v8[7], which includes 17382 samples from 980 healthy individuals, encompassing a variety of tissues. Analyses were performed in R statistical environment (v4.2.3) and sequencing counts were normalized and further processed using the variable stabilizing transformation (VST) algorithm embedded within the *DESeq2* (v1.38.3) package[34]. We focused at removing batch effects prior to performing tissue-specific correlation to age. Batch effects were explored based on Principal Component analyses (PCA), which were performed separately for every tissue (the top 1000 most variable genes were used for the dimensionality reduction). The age distributions were consistent across tissues, as the multiple samples from different tissues originated from the same donors, all falling within the age range of 20–70 years. Given that gender and the precise

anatomical sub-localization within the same tissue-type (i.e., Brain region) were the most prominent sources of variability, data were corrected for these categorical variables using the *removeBatchEffect* algorithm, included in the R package *limma* (v3.54.2)[35]. Importantly, when adjusting for these features, we defined the chronological age of the donor as the batch covariate in the design matrix, making thereby sure that age-related differences would not be biased by batch corrections. To ensure robust correlation results between gene expression and age, we implemented a multi-step resampling and outlier removal procedure. This workflow was designed to minimize the influence of outliers and ensure reliable estimates across tissues. Specifically, we employed a 10-fold Leave-Half-Out (LHO) resampling approach, where the dataset for each tissue was randomly divided into 10 subsets, each containing 50% of the original samples. For each subset, Spearman's rank correlations were computed between gene expression and age. Before estimating the correlations, outliers were identified and excluded if their expression values exceeded 3 standard deviations from the mean for each gene. This process was repeated for all 10 subsets, and the resulting correlation estimates were averaged to provide a robust estimate of the gene-age correlation for each tissue. Subsequently, mean correlations across tissues were calculated to provide an aggregate result. This resampling procedure allowed us to account for potential biases and ensure the robustness of our findings. In parallel, for completeness, we also computed Pearson's correlations between gene expression and age without applying the resampling approach: for each tissue individually, we computed Pearson's correlations between gene expression and age (represented in years) for all available samples. We derived p-values associated to correlation coefficients and, for each tissue considered, we adjusted p-values for multiple testing using the false discovery rate (FDR) method. To rigorously evaluate the statistical significance of the observed correlation between *EDA2R* expression and age in multiple human tissues, we conducted permutation tests ($n = 1000$) for each tissue, a well-established method for assessing the likelihood of random associations. In each permutation, we reshuffled the age of donors and then we recalculated Pearson's correlation between the randomized age and gene expression values. To determine the statistical significance of the observed correlation, we compared the Pearson's r values derived from the actual data to the distribution of r values obtained from the permutations. We calculated a p-value, which represents the proportion of simulated correlations that were equal to or greater than the correlation we observed in the real data.

The muscle biopsy study (FUSION) was approved by the Hospital District of Helsinki and Uusimaa (this data was used with permission; Database of Genotypes and Phenotypes (dbGaP) Study Accession: phs001048.v2.p1) and involved 291 Finnish individuals (166 men, age 59.5 ± 8.1 years; 125 women, age 60.3 ± 8.1 years), as previously described[36]. Briefly, muscle samples were obtained from the m. vastus lateralis using a conchotome, under local anesthesia, with 20 mg·ml⁻¹ lidocaine hydrochloride without epinephrine. Expression of the *EDA2R* gene was calculated using htseq-count v0.5.4[37] with the basic GEN-CODE v19 annotation as reference transcriptome[38] and presented in transcripts per kilobase million (TPM). The analysis of high-throughput experiments involving human myoblasts was performed by mapping sequencing read to the human genome (GRCh38) with STAR[39] aligner and Gencode[40] annotations (v38). Processed single-cell sequencing data of human skeletal muscles were retrieved from the Human Muscle Ageing Cell Atlas[17] (HLMA, https://db.cngb.org/cdcp/hlma).

## Analysis workflow for murine samples

Gene expression data derived from multiple tissues of mice (adipose tissue, brain, liver, lung, limb, small intestine, spleen, pancreas, heart, kidney, skin) and rats (adrenal gland, brain, heart, kidney, lung, skeletal muscle, spleen, thymus, testes, pancreas, uterus) were retrieved from GSE132040 and GSE53960, respectively. We grouped and processed together adipose tissue samples originating from brown, gonadal, mesenteric and subcutaneous adipose tissues. To account for varying numbers of replicates at different time points in mice, we calculated median gene expression values across samples. Sequencing counts were normalized and further processed as described for human samples. Microarray-derived gene expression data (Affymetrix Mouse Genome 430 2.0 Array) for the gastrocnemius muscle of young (10 weeks) and old (24 months) mice were obtained from GSE52550. Data were reprocessed from raw files using the *affy* (v1.76.0) package[41] and normalized with the Robust Multi-Array expression measure using sequence information provided by the *gcrma* package[42]. Downstream differential expression was performed through *limma*. RNA-Seq data for the gastrocnemius muscle of rats were retrieved from SRA project PRJNA516151. Differential expression results of all genes are provided by the authors and are accessible in from of supplementary files from the respective publication[8]. Probeset-level analysis of microarray experiments of murine models of HPGS syndrome involving kerati-nocytes (GSE67288)[9] and gonadal adipocytes (GSE51203)[10] were process in R Statistical environment using the limma package[35]. For the analysis of high-throughput experiments, sequencing reads generated for murine myoblasts and myotubes were mapped to the reference mouse genome (GRCm39) with STAR[39] aligner and Gencode[40] annotations (M27). Raw counts where then processed in R Statistical environment and downstream analysis was performed through the DESeq2 pipeline. RNA-Seq data for LMNA-G609G murine models of Hutchinson-Gilford Progeria Syndrome (HGPS) were retrieved from GSE165409 and mapped to the reference mouse genome (GRCm39) using STAR aligner and Gencode annotations (M27). Processed single cell sequencing data of mouse skeletal muscles were retrieved from Dryad repository (10.5061/dryad.t4b8gtj34)[14] and from Zhang et al.[15] (https://mayoxz.shinyapps.io/Muscle). In the dataset from McKellar et al.[14], UMAP representations were precomputed based on principal component analysis (PCA) for dimensionality reduction. Additionally, Harmony batch correction was applied to account for batch effects across diverse datasets, including different ages, injury models, and experimental conditions[14]. These preprocessed UMAP representations were utilized directly in our analysis to ensure alignment with the original data structure. For the Zhang et al. dataset[15], a standard UMAP representation was used, based on principal component analysis (PCA) without additional batch correction. The processed data and UMAP representations were sourced directly from the publicly available repository and utilized in our analysis.

## Gene-set enrichment analysis

Gene-set enrichment analysis used for dotplot representations was performed with *Camera*[43] (limma v3.54.2) using Reactome (v77) and Hallmark gene-sets (MSigDB v7) collections. Input genes were ranked based on the adjusted p-value resulting from either correlation testing or differential expression analysis.

Pathway enrichment maps (FDR < 0.1, overlap coefficient = 0.5) were generated on top of Gene Set Enrichment Analysis results (GSEA v4.3.2) and represented using Cytoscape (v3.9.1). To this purpose multiple gene sets collections were tested for significance for mouse and humans:

- mh.all.v2023.2.Mm (mouse); h.all.v2023.2.Hs (human)
- m2.cp.reactome.v2023.2.Mm (mouse); c2.cp.reactome.v2023.2.Hs (human)
- m2.cp.wikipathways.v2023.2.Mm (mouse); c2.cp.wikipathways.v2023.2.Hs (human)
- m5.go.bp.v2023.2.Mm (mouse); c5.go.bp.v2023.2.Hs (human)

## Subjects, blood sampling and RNA extraction (NTR, NESDA cohorts)

Two parent projects that supplied data for this study are large-scale longitudinal studies: the Netherlands Twin Registry (NTR)[43] and the Netherlands Study of Depression and Anxiety (NESDA)[44]. NTR and NESDA studies were approved by the Central Ethics Committee on Research Involving Human Subjects of the VU University Medical Center, Amsterdam (IRB number IRB-2991 under Federalwide Assurance 3703; IRB/institute codes, NESDA 03-183; NTR 03-180), and all subjects provided written informed consent. The sample consisted of 5228 subjects (before QC), 3164 participants from NTR (2668 twins, 21 multiples, 284 siblings from these twins and 187 parents, and 4 spouses) and 2064 unrelated participants from NESDA. The age of the participants ranged from 17 to 79 years (mean 37.5, SD 13) and 65% of the sample was female. As part of the NESDA and NTR biobank protocols, data on weight, standing height, CRP and current smoking (yes/no) were collected in all participants.

The NTR and NESDA blood sampling and RNA extraction procedures have been described in detail previously. In short, for NTR, venous blood samples were drawn between 0700-1100 after an overnight fast and usually in the subjects' homes. Within 20 min of sampling, heparinized whole blood was transferred into PAXgene Blood RNA tubes (Qiagen) and stored at −20 °C. The PAXgene tubes were shipped to the Rutgers University Cell and DNA Repository (RUCDR), USA. Average time between blood sampling and RNA extraction was 211 weeks (included in mixed model for gene expression). Upon registration of samples, RNA was extracted using Qiagen Universal liquid handling system (PAXgene extraction kits as per the manufacturer's protocol). From the NESDA subjects, serial venous whole blood samples were obtained (8–10 AM, after overnight fasting) in one 7-mL heparin-coated tube (Greiner Bio-One, Monroe, North Carolina). Between 10 and 60 min after blood draw, 2.5 mL of blood was transferred into a PAXgene tube (Qiagen, Valencia, California). This tube was kept at room temperature for a minimum of 2 h and then stored at −20 °C. Average time between blood sampling and RNA extraction was 113 weeks (included in mixed model for gene expression). Total RNA was extracted at the VU University Medical Center (Amsterdam) according to the manufacturer's protocol (Qiagen). For both NESDA and NTR samples high molecular weight genomic DNA was isolated from frozen blood in EDTA tubes using Puregene DNA isolation kits (Qiagen).

## Gene expression measurements and quality control (NTR, NESDA cohorts)

RNA quality and quantity was assessed by Caliper AMS90 with HT DNA5K/RNA LabChips. RNA samples that showed abnormal ribosomal

subunits in the electropherograms were removed. Genome-wide gene expression assays were conducted at the Rutgers University Cell and DNA Repository (RUCDR, http://www.rucdr.org)[45]. NTR and NESDA samples were randomly assigned to plates with seven plates containing subjects from both studies to better inform array QC and study comparability. For cDNA synthesis, 50 ng of RNA was reverse-transcribed and amplified in a plate format on a Biomek FX liquid handling robot (Beckman Coulter) using Ovation Pico WTA reagents per the manufacturer's protocol (NuGEN). Products purified from single primer isothermal amplification (SPIA) were then fragmented and labeled with biotin using Encore Biotin Module (NuGEN). Prior to hybridization, the labeled cDNA was analyzed using electrophoresis to verify the appropriate size distribution (Caliper AMS90 with a HT DNA 5 K/RNA LabChip). Samples were hybridized to Affymetrix U219 array plates (GeneTitan) to enable highthroughput gene expression profiling of 96 samples at once.

The U219 array contains 530,467 probes for 49,293 transcripts. All probes are 25 bases in length and designed to be "perfect match" complements to a designated transcript. Array hybridization, washing, staining, and scanning were carried out in an Affymetrix GeneTitan System per the manufacturer's protocol. Probes were removed when their location was uncertain or if their location intersected a polymorphic SNP (dropped if the probe oligonucleotide sequence did not map uniquely to hg19 or if the probe contained a polymorphic SNP based on HapMap3 and 1000 Genomes project data). Expression values were obtained using RMA normalization implemented in Affymetrix Power Tools (APT, v 1.12.0). First, 103 samples with array results inconsistent with the phenotypic database were removed (inconsistent sex based on chr X and chr Y probe sets). Second, we used the pairwise correlation matrix of expression profiles across all arrays for additional QC. These quantities were expressed in terms of median absolute deviations to provide a sense of scale. We used:

$$D_i = \frac{|r_i - r|}{median(|r_k - r|)_{\{k=1...N\}}} \tag{1}$$

With $r_i$ the average of correlations for sample $i$, and $r$ the average of all correlations. Larger values of D corresponded to poor quality; 80 samples with $D > 5$ were removed, decreasing the final number of subjects to 5362. Out of the 49,293 transcripts, 2 could be assigned to the EDAR2 gene. The average expression values of these 2 transcripts, obtained using RMA normalization, were used in the association analyses.

## Physical activity assessment and associations to *EDA2R* expression

In NESDA, Physical Activity[46] was assessed using the International Physical Activity Questionnaire (IPAQ)[47] and expressed as overall energy expenditure in Metabolic Equivalent Total (MET) minutes per week (MET level * minutes of activity * events per week). In NTR, Physical Activity was assessed as regular voluntary exercise behavior using the mean of survey based self-reports across up to 8 repeated measurements across 2-yearly data collection waves[48,49]. Participants were first asked whether they regularly participate in exercise in their leisure time ("Yes" or "No"). If the response was affirmative they were asked (1) which sport they participate in, (2) how many years they have been doing so, (3) how many months a year they do so, (4) how many times a week on average they do so, (5) how many minutes on each occasion they exercise on average. Metabolic equivalent of task (MET) values were assigned to each exercise activity following Ainsworth et al.[50]. This MET value reflects the energy expenditure of the type of exercise as a multiple of basal energy expenditure (approximately 1 kcal/ kg/h). Using these data, all exercise activities were assigned a weekly MET-minutes value by calculating the product of (1) the intensity as the MET value, (2) the weekly frequency, and (3) the

average duration in minutes. Per survey, the total volume of exercise behavior of the participant (METmin) was the sum of the weekly METminutes across all exercise activities. Obligatory exercise such as physical education (PE) classes, or other non-voluntary activity such as biking as a form of transportation, as well as seasonal exercise, such as skiing during winter only, were excluded, as they are not part of regular voluntary exercise behavior. The mean METmin score across 4.2 (range 1–8) surveys was used, but analyses were also repeated using the survey closest (<7 months) to the moment of participation in the Biobank study collecting the mRNA samples. Because of the conceptual difference in the assessments of physical activity in NESDA and NTR we ran two separate analyses on the association of PA on the EDA2R expression. In NESDA, linear mixed models (package *lm* in R) were used with a transformation of EDA2R gene expression (adjusted for plate effects) as dependent variable (log(1 + EDA2R)). Independent model covariates were selected based on significance of the variable in the fitted mixed models. Fixed effect covariates included in the final model were sex, age, body mass index (BMI, weight/height 2 in kg/m), smoking status (yes/no current smoking), and log(CRP). In NTR generalized equation estimation models (package gee) were used with a transformation of EDA2R gene expression as dependent variable (log(1 + EDA2R)). Independent model covariates were selected based on significance of the variable in the fitted mixed models. Fixed effect covariates included in the final model were sex, age, body mass index (BMI, weight/height 2 in kg/m), smoking status (yes/no current smoking), and log(CRP). Random effect was family ID.

## Reporting summary
Further information on research design is available in the Nature Portfolio Reporting Summary linked to this article.

## Data availability
The raw sequencing data generated in this study have been deposited in the EBI-Biostudies database under accession codes E-MTAB-13101, E-MTAB-14234, and E-MTAB-14239. GTEx v8 RNA-Seq data is available in the Database of Genotypes and Phenotypes (dbGap) database under the accession code phs000424 [https://www.ncbi.nlm.nih.gov/projects/gap/cgi-bin/study.cgi?study_id=phs000424.v10.p2], and publicly avaiable raw counts were downloaded from the GTEx Portal [https://gtexportal.org]. Data from the muscle biopsy study (FUSION) were used with permission and are available in dbGaP database under the accesion phs001048.v2.p1 [https://www.ncbi.nlm.nih.gov/projects/gap/cgi-bin/study.cgi?study_id=phs001048.v2.p1]. Processed single-cell sequencing data of human skeletal muscles were downloaded from the Human Muscle Ageing Cell Atlas [https://db.cngb.org/cdcp/hlma]. Processed single-cell sequencing data of mouse skeletal muscles used are available in the Dryad repository under the accession code 10.5061/dryad.t4b8gtj34 [https://datadryad.org/stash/dataset/doi:10.5061/dryad.t4b8gtj34] and from [https://mayoxz.shinyapps.io/Muscle]. Gene expression data derived from multiple tissues of mice (adipose tissue, brain, liver, lung, limb, small intestine, spleen, pancreas, heart, kidney, skin) and rats (adrenal gland, brain, heart, kidney, lung, skeletal muscle, spleen, thymus, testes, pancreas, uterus) are available in GEO database under the accession codes GSE132040 and GSE53960. Microarray-derived gene expression data (Affymetrix Mouse Genome 430 2.0 Array) for the gastrocnemius muscle of young (10 weeks) and old (24 months) mice are available in the GEO database under the accession code GSE52550. Microarray experiments of murine models of HPGS syndrome involving keratinocytes and gonadal adipocytes are available in the GEO database under the accession codes GSE67288 and GSE51203. RNA-Seq data for the gastrocnemius muscle of rats is available in the SRA database under the accession code PRJNA516151. Microarray gene expression of blood samples from the NESDA and NTR studies was used with permission and is available in dbGaP databased under the accession code phs000486 [https://

| Accession | Data source | Species | Data type | Organ/cell type | Age | Analysis |
|---|---|---|---|---|---|---|
| E-MTAB-13101 | generated for this study | *Mus musculus* | RNA-Seq | C2C12 myoblasts | 2-month-old C3H mouse | Overexpression of *Eda2r* |
| E-MTAB-14234 | generated for this study | *Mus musculus* | RNA-Seq | C2C12 myoblasts differentiated into myotubes | 2-month-old C3H mouse | Overexpression of *Eda2r* |
| E-MTAB-14239 | generated for this study | *Homo sapiens* | RNA-Seq | myoblasts | 65 years | Overexpression of *EDA2R* Silencing of *EDA2R* Treatment with EDA-A2 |
| phs000424.v8 | GTEX v8 | *Homo sapiens* | RNA-Seq | adipose tissue, adrenal gland, blood, brain, breast, colon, esophagus, heart, kidney, liver, lung, muscle, peripheral nerve, ovary, pancreas, pituitary gland, prostate, salivary gland, skin, small intestine, spleen, stomach, testis, thryoid, uterus, vagina, blood vessels | 20-70 years | Correlations between gene expression and age |
| GSE132040 | Tabula Muris Senis | *Mus musculus* | RNA-Seq | adipose tissue, brain, liver, lung, limb, small intestine, spleen, pancreas, heart, kidney, skin | 1-27 months | Correlations between gene expression and age |
| GSE53960 | Rat BodyMap | *Rattus norvegicus* | RNA-Seq | adrenal gland, brain, heart, kidney, lung, skeletal muscle, spleen, thymus, testes, pancreas, uterus | 2-104 weeks | Correlations between gene expression and age |
| GSE165409 | LMNA-G609G HGPS | *Mus musculus* | RNA-Seq | descending aorta | young | HPGS mice vs age-matched wild-type |
| GSE67288 | LMNA-G609G HGPS | *Mus musculus* | gene expression microarray | keratinocytes | 24-35 days postnatal | HPGS mice vs age-matched wild-type |
| GSE51203 | LMNA-G609G HGPS | *Mus musculus* | Gene expression microarrays | adipocytes | 4 months | HPGS mice vs age-matched wild-type |
| phs001048 | FUSION Tissue Biopsy Study | *Homo sapiens* | RNA-Seq | vastus lateralis muscle | 20-79 years | *EDA2R* expression in aging muscle |
| GSE52550 | Young and old mice | *Mus musculus* | Gene expression microarrays | Gastrocnemius muscle | 10 weeks-24months | *EDA2R* expression in aging muscle |
| PRJNA516151 | Rats of different ages | *Rattus norvegicus* | RNA-Seq | Gastrocnemius muscle | 6-27 months | *EDA2R* expression in aging muscle |
| 10.5061/dryad.t4b8gtj34 (Dyrad) | integration of single cell data | *Mus musculus* | scRNA-Seq | skeletal muscle | 4-20 months | *EDA2R* expression at single cell resolution |
| CNP0004394, CNP0004395, CNP0004494, CNP0004495 | Human Muscle Ageing Cell Atlas 10 (HLMA, https://db.cngb.org/cdcp/hlma | *Homo sapiens* | scRNA-Seq | muscle | 15-99 years | *EDA2R* expression at single cell resolution |
| GSE172410 | https://mayoxz.shinyapps.io/Muscle | *Mus musculus* | scRNA-Seq | muscle | 6-24 months | *EDA2R* expression at single cell resolution |
| phs000486 | Netherlands Study of Depression and Anxiety (NEDA) Netherlands Twin Registry (NTR) | *Homo sapiens* | Gene expression microarrays | blood | 18-65 years | Association between *EDA2R* and physiological parameters |

www.ncbi.nlm.nih.gov/projects/gap/cgi-bin/study.cgi?study_id=phs000486.v1.p1]. A source file containing the numeric data used to generate the figure panels is deposited along with the manuscript. In addition, a comprehensive list of all datasets used in the study is provided to facilitate reproducibility.

Source data are provided with this paper.

## Code availability
The code to easily reproduce our analyses for human subjects, animal models, and experimental data, which relies on existing algorithms and is based on established methodologies, is available on our GitHub repository: https://github.com/BolisLab/eda2r and in Zenodo repository 10.5281/zenodo.14279442). Additionally, a Source Data file has been deposited to facilitate the reproduction of individual figure panels.

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

## Acknowledgements

This work was supported in part by the Giovagnoni Bequest, which provided financial support that contributed to the research and its completion.

## Author contributions

M.B., MC.B., S.N.M. and R.P., originally developed the concept and further elaborated on it. M.B., L.G., GA.C., A.V., L.D.R., carried out bioinformatic analyses, B.S. provided human myoblasts for revision and M.L., A.D.R.C. performed experiments. F.C. extended analyses to larger datasets under the supervision of L.G., EJC.G, B.P., J.D. R.J. performed analyses and assessed correlations on NESDA and NTR cohorts, MC.B., A.D.R.C., M.R. and I.C. performed functional experiments, I.C. and MC.B. prepared RNA-Seq libraries, I.I.A., E.V.G. and Molecular Genetics Group, provided transcriptomic data, performed analyses and statistical associations involving human samples from vastus lateralis muscle, M.B., R.P., S.N.M., GA.C., wrote/revised the manuscript and designed the figures.

## Competing interests

The authors declare no competing interests.

## Additional information

[1]Computational Oncology Unit, Department of Oncology, Istituto di Ricerche Farmacologiche 'Mario Negri' IRCCS, Via Mario Negri 2, 20156 Milano, Italy. [2]Department of Biotechnology and Biosciences, University of Milano-Bicocca, Milano, Italy. [3]Laboratory of Muscle Pathophysiology, Department of Neuroscience, Istituto di Ricerche Farmacologiche 'Mario Negri' IRCCS, Via Mario Negri 2, 20156 Milano, Italy. [4]Institute of Oncology Research, Bellinzona, Switzerland. [5]Università Della Svizzera Italiana (USI), Faculty of Biomedical Sciences, Bellinzona, Switzerland. [6]Department of Biosciences, University of Milan, Via Celoria 26, 20133 Milan, Italy. [7]Laboratory of Human Pathology in Model Organism, Department of Molecular Biochemistry and Pharmacology, Istituto di Ricerche Farmacologiche 'Mario Negri' IRCCS, Via Mario Negri 2, 20156 Milano, Italy. [8]Department of Research in Inflammation and Immunology, IRCCS Humanitas Research Hospital, Milan, Italy. [9]Friedrich-Baur-Institute, Department of Neurology, LMU Klinikum, Ludwig-Maximilians University, Munich, Germany. [10]Department of Molecular Biology and Genetics, Lopukhin Federal Research and Clinical Center of Physical-Chemical Medicine of Federal Medical Biological Agency, Moscow, Russia. [11]Department of Psychiatry, Amsterdam UMC location Vrije Universiteit Amsterdam, Amsterdam, the Netherlands. [12]Amsterdam Public Health, Mental Health Program, Amsterdam, The Netherlands. [13]Amsterdam Neuroscience, Mood, Anxiety, Psychosis, Sleep & Stress Program, Amsterdam, The Netherlands. [14]Department of Biological Psychology, Vrije Universiteit Amsterdam, Amsterdam, The Netherlands. [15]Research Institute for Sport and Exercise Sciences, Liverpool John Moores University, Liverpool L3 5AF, UK. [16]Department of Physical Education, Plekhanov Russian University of Economics, Moscow, Russia. [17]Laboratory of Genetics of Aging and Longevity, Kazan State Medical University, Kazan, Russia. [18]Swiss Institute of Bioinformatics, Bioinformatics Core Unit, Bellinzona TI 6500, Switzerland. [20]These authors contributed equally: Maria Chiara Barbera, Luca Guarrera, Andrea David Re Cecconi. [21]These authors jointly supervised this work: Ildus I.Ahmetov, Marco Bolis. ✉e-mail: marco.bolis@ior.usi.ch; marco.bolis@marionegri.it

## Molecular Genetics Group

Rinat I. Sultanov[10], Alexandra Kanygina[10], Nikolay A. Kulemin[10] & Ekaterina A. Semenova[10,19]

[19]Research Institute of Physical Culture and Sport, Volga Region State University of Physical Culture, Sport and Tourism, Kazan, Russia.

