## [Transparent Peer Review file · Nature Communications]

Increased Ectodysplasin-A2-Receptor EDA2R is a Ubiquitous Hallmark of Aging and Mediates Parainflammatory Responses

Corresponding Author: Dr Marco Bolis

Version 0:

Reviewer comments:

Reviewer #1

(Remarks to the Author)

This paper reports on an aging-associated increase of the transmembrane Ectodysplasin-A2-Receptor (EDA2R) across various tissues in humans and in animal models including in Hutchinson-Gilford progeria mouse model of accelerated aging. Overexpression of *Eda2r* in undifferentiated murine myoblasts increased proinflammatory signaling and inflammation-related signaling pathway. Authors suggest that EDA2R is a promising therapeutic target for aging-associated pathologies.

Critique:

The present study largely used a data mining approach and more rigorously documents the association of increased EDA2R expression with aging. The bioinformatics and statistical approaches are appropriate.

Prior studies already demonstrated aging-associated increases of EDA2R in certain tissues and that it mediates proinflammatory responses. A role of EDA2R promoting sarcopenia has been reported previously in cancer and other settings and been reviewed recently (PMID: 37596181).

The function studies are very limited to EDA2R overexpression in cells and RNA-sequencing. These experiments are lacking details about sample numbers.

Conclusion: The main weaknesses of the present study are its narrow scope, and limited originality.

Reviewer #2

(Remarks to the Author)

In this manuscript, Barbera et al investigated age-associated changes in EDA2R expression. Analyzing datasets available at GTEX and elsewhere, the authors demonstrated that EDA2R mRNA levels correlate with increasing age in various tissues, including skeletal muscle. A similar association was also detected at the protein level when a dataset on plasma proteome was analyzed. EDA2R expression increased particularly in skeletal muscle with age in humans, mice and rats. The analysis of a single-cell RNA-sequencing dataset indicated high EDA2R transcription in myogenic precursors, arguing for a role in muscle development and repair. Functional experiments revealed that the overexpression of EDA2R induces transcription of pro-inflammatory factors in cultured myoblasts, resembling alterations seen in aging muscle. Large-scale transcriptomics data from human cohorts confirmed the correlation between EDA2R and age and a positive association with circulating C-reactive protein, a marker of systemic inflammation. The study suggests that targeting EDA2R could be a novel approach to mitigate age-associated phenotypes.

I believe this is a well-written manuscript, which provides a comprehensive analysis of aging-related gene expression patterns and reports EDA2R as a prominent marker for aging. Recent studies also linked EDA2R to muscle wasting

associated with cancer cachexia and demonstrated EDA2R upregulation in muscle tissue upon aging. Some key studies were not cited in this manuscript. I think the authors should expand their discussion by referring these publications as described below. Additionally, it would be important for the authors to show or discuss any correlations between EDA2R and sarcopenia. My concerns and suggestions are listed as follows:

1. Whether EDA (particularly EDA-A2) expression correlates with aging is an important point. This data was shared in supplementary figure 4 but it was not discussed in the main text. A correlation between blood EDA/EDA-A2 mRNA and aging was shown Figure 2i-j. I believe the discussion on the EDA2R ligand should be expanded. In fact, changing EDA protein levels were previously described by Lehallier et al (PMID: 31806903). In this paper, plasma EDA2R protein expression was reported to be positively correlated with aging in humans and mice whereas EDA expression was up- and down- regulated in human and mouse samples, respectively. The authors should cite this work and refer to these findings in their discussion.
2. The authors indicated that EDA2R protein abundance was increased in multiple tissues with age (line 90). The results were shared in supplementary figure 5, which refers to a dataset originally reported by Lehallier et al. but also expanded more recently. It is not clear if these results represent EDA2R protein expression in multiple tissues or just plasma samples. The results in this figure can be better described. If the data points correspond to different tissues, then a separate graph for each can be provided. Would it be possible to do a similar analysis for EDA/EDA-A2? This would strengthen the findings on EDA mRNA reported in Fig 2i-j.
3. The authors demonstrated that EDA2R was upregulated in the aortic artery of the HGPS mouse model. The significance of this observation made in the aortic artery should be discussed in the main text. Would it be possible to test EDA2R expression in other tissues of HGPS mice using additional datasets?
4. The authors performed a high-resolution analysis of EDA2R expression in muscle tissue of mice using a dataset of a single-cell RNA-sequencing study and reported high EDA2R expression in satellite cells. However, how EDA2R expression in these cells changes upon aging was not reported. There is at least one single-cell RNA-sequencing study comparing young and old muscle samples: Zhang et al (PMID: 36147777). In fact, this paper reported (using bulk RNA-sequencing) increased EDA2R expression in muscle samples from old humans and old mice (Figure 6a and 7i). This work should be cited.
5. While single-cell RNA sequencing is used to study gene expression in mono-nucleated cells of muscle tissue, it is not ideal for studying myonuclei of multinucleated muscle fibers, which are underrepresented due to cell isolation. Instead, single-nucleus RNA-sequencing analysis of muscle samples would better represent gene expression in muscle fibers. In fact, age-dependent EDA2R expression has already been reported in a study utilizing single-cell RNA-sequencing. Perez et al (PMID: 36516485) demonstrated that EDA2R transcript was induced significantly in muscle biopsies of elderly subjects compared with young controls and there was a trend for increased EDA2R expression in the sarcopenic subjects. Furthermore, this study also identified a population of EDA2R+ myonuclei only present in the fast muscle fibers of older adults. This work should be cited as well.
6. In Figure 2e, an EDA2R plasmid was introduced into C2C12 myoblasts. However, EDA2R overexpression is very mild (< 2 fold). Therefore, gene expression changes may not reflect the full potential of the effect of EDA2R on the myoblast transcriptome. The authors may consider viral gene delivery. A similar experiment may also be considered in differentiated myotubes.
7. Any correlation between EDA2R expression and sarcopenia in the elderly was not discussed. It is hard to find datasets with phenotypic parameters (i.e. muscle mass or physical performance), however, such results would significantly enhance the novelty of the manuscript.
8. Please identify explicitly the human dataset used in the gene set enrichment analysis of Figure 2g.
9. Please correct the typos in line 111 and 196.

Reviewer #3

(Remarks to the Author)

The study by Barbera et al identifies a strong association between age and increased transcript levels of EDA2R, the receptor for the EDA2 isoform of Ectodysplasin A. This is observed in virtually all tissues in human, mice and rats. EDA2R transcripts are also increased in a mouse model of accelerated ageing. The authors further study the association of EDA2R transcripts with age in muscles and blood. In muscles, the correlation is confirmed. In human blood, EDA2R is apparently not as strongly increased as in other tissues, but transcripts levels of EDA, especially those of the EDA2 splice variant, the ligand of EDA2R, increase with age.

Further mining of datasets indicate correlation of EDA2R transcript expression with CRP, an inflammatory marker, and the authors suggest that EDA2R might be linked to an age-related inflammatory status, that they experimentally support by overexpression of EDA2R in a cell line and by measuring induction of inflammation-related genes (e.g IL6, IL1a).

Authors seem to have followed rigorous protocols to maximize reliability in sample comparison. The observation that EDA2R is increased with age is amply supported by data and is very interesting, especially in the context of EDA2R being implicated in muscle loss. The correlation with an inflammatory status is also interesting, although the experimental "proof" that EDA2R contributes significantly to inflammation by overexpressing EDA2R in a cell line is rather weak: EDA2R is not the only inducer of NF- κ B. It is probable that several other members of the TNFR family (to which EDA2R belongs) would

have induced similar genes and pathways upon overexpression, all of which have been described to respond to NF- κ B activation.

I would like to raise two points.

1.- As mentioned in the manuscript, the ligand EDA exists in at least two splice variants: EDA2 that binds EDA2R, and EDA1 that binds EDAR. Given the age-related increase of EDA transcripts in blood (Fig 2e), it would be interesting to indicate transcript levels of EDA in other tissues also, to know whether upregulation of EDA of Fig 2e is a tissue-specific or a general effect. EDAR could also be shown as a control (which I assume will not appear in top transcripts associated with ageing). This should be done at least for Fig 1a.

2.- The source of EDA transcripts in blood is not immediately obvious. It would be an interesting addition to the manuscript if the authors could analyse a data set comparing human blood cell types and pinpoint the potential source of EDA transcripts in blood. As this experiment will not change the conclusion that EDA2R increases with age, I do not insist to have it done, should it present difficulties.

Minor points

a) Fig 1b. The authors have shown names of 6 age-associated transcripts in addition to EDA2R. They could also identify the most downregulated transcript.

b) Methods, line 243. Please indicate the % of FBS used to culture C2C12 cells.

c) The last sentence of the main text suggests that EDA2 - EDA2R could be targeted to counteract ageing. Whether this will work remains highly speculative. The cautious way in which this sentence is written is appropriate, but a stronger statement should not be made.

Pascal Schneider

Reviewer #4

(Remarks to the Author)

The study presents a comprehensive investigation into the role of Ectodysplasin-A2-Receptor (EDA2R) in ageing, identifying it as a tissue-independent biomarker associated with ageing across different species. Through extensive bioinformatics analysis of various datasets and experimental validation, the authors conclude that EDA2R expression increases with age and is linked to parainflammatory responses and sarcopenia-like phenotypes. Targeting EDA2R signaling is therefore proposed as a novel approach to mitigating age-related physiological changes.

Suggestions for Improvement or Clarification:

1. Methodological Transparency: while the study uses a rigorous multi-step bioinformatics approach, the statistical methods and criteria for data filtering (beyond FDR adjustment) could be better detailed to enhance reproducibility (i.e. expanding the information given in lines 261 to 298, maybe in supplementary).
2. Mechanistic Insights: EDA2R's role in ageing is explored extensively, but the molecular mechanisms underlying EDA2R-induced parainflammatory responses could be further elucidated, potentially through pathway analysis and interaction studies. In particular lines 128 to 130 could be expanded a little and supported with data or literature.
3. Therapeutic Implications: the potential for targeting EDA2R as a therapeutic strategy is mentioned and features and one of the most important conclusions of this work; however, while the authors provide clear support for the effects of overexpressing EDA2R, the study could benefit from either preliminary data on pharmacological inhibition or other interventions to lower EDA2R expression in ageing models to substantiate these claims.

Minor Typos or Elements to address:

1. Consistency in Gene/Protein Naming: Ensure consistency in the use of gene and protein names (e.g., EDA2R vs. Eda2r) throughout the manuscript for clarity, making sure the two are indeed used when referring to gene (transcription, expression) or protein (protein levels, signalling).
2. Statistical Details: Provide more detailed statistical analysis information, including exact p-values and n numbers, especially in figure legends and the results description.

Version 1:

Reviewer comments:

Reviewer #1

(Remarks to the Author)

Reviewer #2

(Remarks to the Author)

The authors have satisfactorily addressed all of my comments. This is very much appreciated. I only recommend the following correction regarding the legends of the supplementary figures 6 and 7.

Regarding my previous comment #2 on the supplementary figure 5 (now supplementary figure 6), the authors indicated that they revised the figure legend. The legend provided in the rebuttal is in fact corrected. However, the legend in the main text remains the same as the previous version. It reads as measurements were taken from multiple tissues while it should have been blood plasma only. This correction also applies to the new supplementary figure 7, where EDA expression was measured.

Serkan Kir

Reviewer #3

(Remarks to the Author)

The authors have adequately and thoroughly taken into consideration my comments/suggestions, among others by showing expression levels of EDA and EDAR in comparison to XEDAR, by investigating the possible source of EDA transcripts in blood, and by improving parts of the method section. Thank you. Pascal Schneider

Reviewer #5

(Remarks to the Author)

This is an interesting report and the authors did a good job at leveraging data in the public domain with experimental data that they generated for this study. The original comments of reviewer 4 were well addressed. However, I found some remaining issues with the analysis that could be addressed.

Fig1a: Showing the correlation between age and gene expression is not sufficient because the results could be biased by outliers or funny patterns, so supplement figure 2 is really critical. This figure actually shows that the claim of an increase with age pattern across all tissues is not precise. The scatter plots for brain, breast, liver, pituitary gland for example show that some of the correlation values may be driven by outliers. Did the authors remove outliers/bad samples before conducting the analysis?

Related to this figure, the sentence between lines 487-489 needs clarification. What is the "biological age" they refer to?

Fig1b: why using the median correlation across tissues and not the mean?

Fig 1: The legend of figure 1 should mention sample sizes used for the analysis.

Fig1a: the sample size in the 20-40 age group is very small, so the robustness of the results is questionable. In addition, the data show clear outliers that may drive the significance levels. I would suggest conducting an analysis in which outliers are removed and maybe the age groups 20-40 and 40-50 merged into one.

Fig 2d: I could not find the details of the scRNAseq data. The authors should provide some details of the analysis to show UMAP plots.

Fig 2j: In addition to p-values, it would be useful to show effects in a standardized scale, so that they are comparable.

Version 2:

Reviewer comments:

Reviewer #5

(Remarks to the Author)

The authors have revised the manuscript to address all the issues that I raised and the revised manuscript is acceptable.

REBUTTAL – POINT BY POINT REPLY TO REVIEWERS

Reviewer #1 (Remarks to the Author):

This paper reports on an aging-associated increase of the transmembrane Ectodysplasin-A2-Receptor (EDA2R) across various tissues in humans and in animal models including in Hutchinson-Gilford progeria mouse model of accelerated aging. Overexpression of Eda2r in undifferentiated murine myoblasts increased proinflammatory signaling and inflammation-related signaling pathway. Authors suggest that EDA2R is a promising therapeutic target for aging-associated pathologies.

Critique: The present study largely used a data mining approach and more rigorously documents the association of increased EDA2R expression with aging. The bioinformatics and statistical approaches are appropriate. Prior studies already demonstrated aging-associated increases of EDA2R in certain tissues and that it mediates proinflammatory responses. A role of EDA2R promoting sarcopenia has been reported previously in cancer and other settings and been reviewed recently (PMID: 37596181). Conclusion: The main weaknesses of the present study are its narrow scope, and limited originality.

We respectfully acknowledge the reviewer's critique. However, it is important to emphasize that the reviewer referred to a beautiful review from the Kir. S group (PMID: 37596181) which did cite the pre-print of our manuscript.

Furthermore, the Kir group published one research paper (Nature, 2023 PMID: 37165186) and three reviews/opinions on the topic on high-impact journals (Trends Pharmacol Sci. 2023 PMID: 37596181; Trends Mol Med. 2024 PMID: 38443222; Curr Opin Support Palliat Care 2024; PMID: 38801457) all of which directly cite our manuscript, despite it being in its pre-print form. This strongly underscores the relevance of our research. Moreover, a thorough search of NGS databases such as NCBI (GEO/SRA), and EMBL-EBI (ENA) reveals that there is, as of today, a total of zero deposited experiments involving overexpression of *EDA2R*, except ours. This again reinforces the novelty and originality of our work.

We are aware of the risks associated with pre-printing manuscripts, as it exposes unpublished material to the community. However, this is a choice that prioritizes and benefits scientific dissemination, and we strongly believe that researchers should not be penalized for making this decision.

The function studies are very limited to EDA2R overexpression in cells and RNA-sequencing. These experiments are lacking details about sample numbers.

All figure panels now report in legends the exact number of biological replicates used for the analyses along with the statistical tests performed. To enable complete reproducibility, we generated a source file (SourceData.xlsx) which is now deposited along with the manuscript, containing numeric values used for generating each figure panel. Furthermore, all source code used in the original submission and throughout the revision process is now deposited at our GitHub repository (<https://github.com/BolisLab/eda2r>).

In response to the reviewer's concerns, we have significantly expanded our experimental scope to strengthen our findings.

- We explored whether our results could be extended from myoblast progenitors to terminally differentiated myotubes. RNA-Seq analysis demonstrated that the inflammatory responses observed in progenitors are confirmed in differentiated myotubes (**Figure 2f-h and Supplementary Figure 5f-h**).
- We have successfully translated our findings from mice to humans using myoblasts derived from human donors. These results were validated by transfecting using both lipofectamine and electroporation techniques, confirming the translational potential of our findings. Pathway analysis in this setting revealed that *EDA2R* overexpression strongly associates with signaling pathways related to muscle development and myogenic differentiation (**Figure 2f,i and Supp. Figure 5i**).
- To further validate our findings without exogenous RNA manipulation, we treated human myoblasts with EDA-2 ligands. This treatment induced strong para-inflammatory responses and activation of downstream signaling, mirroring the effects of *EDA2R* overexpression, thereby providing further

rational for developing small molecules aimed at hindering the formation of the EDA2R/EDA-A2 complex (**Figure 2m**).

- We evaluated the effects of *EDA2R* knock-down by performing silencing experiments in human myoblasts. Our results revealed transcriptional perturbations leading to potent downregulation of pro-inflammatory cytokines and other biological pathway linked to parinflammatory responses (**Supplementary Figure 5j**).
- We investigated novel single-cell sequencing data made available by a recent publication from Lai et al. (PMID: 38649488) which includes both single-cell and single-nuclei sequencing, thereby solving the problem related to the underrepresentation of polynucleated cells. We could clearly determine that *EDA2R* expression is increased with age in all cell-types considered, which supports that the increase of *EDA2R* with age is a behavior that can be generalized to different cells of origin (**Figure 2d,e and Supplementary Figure 5b**).

Currently, there is no established antagonist specifically targeting EDA2R, which limits direct pharmacological interventions. We are confident that our manuscript will stimulate further research and interest in developing EDA2R-specific inhibitors, which could provide new therapeutic avenues for managing inflammation and related conditions associated with aging. We agree that further studies, including pharmacological inhibition, would be valuable. Future work could focus on developing and testing potential EDA2R antagonists to complement our findings and substantiate the therapeutic implications of targeting EDA2R in aging models.

Reviewer #2 (Remarks to the Author):

In this manuscript, Barbera et al investigated age-associated changes in EDA2R expression. Analyzing datasets available at GTEX and elsewhere, the authors demonstrated that EDA2R mRNA levels correlate with increasing age in various tissues, including skeletal muscle. A similar association was also detected at the protein level when a dataset on plasma proteome was analyzed. EDA2R expression increased particularly in skeletal muscle with age in humans, mice and rats. The analysis of a single-cell RNA-sequencing dataset indicated high EDA2R transcription in myogenic precursors, arguing for a role in muscle development and repair. Functional experiments revealed that the overexpression of EDA2R induces transcription of pro-inflammatory factors in cultured myoblasts, resembling alterations seen in aging muscle. Large-scale transcriptomics data from human cohorts confirmed the correlation between EDA2R and age and a positive association with circulating C-reactive protein, a marker of systemic inflammation. The study suggests that targeting EDA2R could be a novel approach to mitigate age-associated phenotypes.

I believe this is a well-written manuscript, which provides a comprehensive analysis of aging-related gene expression patterns and reports EDA2R as a prominent marker for aging. Recent studies also linked EDA2R to muscle wasting associated with cancer cachexia and demonstrated EDA2R upregulation in muscle tissue upon aging. Some key studies were not cited in this manuscript. I think the authors should expand their discussion by referring these publications as described below. Additionally, it would be important for the authors to show or discuss any correlations between EDA2R and sarcopenia.

We acknowledge that our manuscript could be enhanced by including critical references and greatly appreciate the reviewer's efforts in directing us to these research papers. We have carefully reviewed the additional literature and agree that the incorporation of these studies strengthen the discussion and provided a more comprehensive context for our findings. The manuscript has been updated to include these references, which now further substantiate our claims and provide a broader overview of the field.

Whether EDA (particularly EDA-A2) expression correlates with aging is an important point. This data was shared **in supplementary figure 4** but it was not discussed in the main text. A correlation between blood **EDA/EDA-A2** mRNA and aging was shown **Figure 2i-j**. I believe the discussion on the EDA2R ligand should be expanded. In fact, changing EDA protein levels were previously described by Lehallier et al (PMID: 31806903). In this paper, plasma EDA2R protein expression was reported to be positively correlated with aging in humans and mice whereas EDA expression was up- and down- regulated in human and mouse samples, respectively. The authors should cite this work and refer to these findings in their discussion. The authors indicated that EDA2R protein abundance was increased in multiple tissues with age (line 90). The results were shared in supplementary figure 5, which refers to a dataset originally reported by Lehallier et al. but also expanded more recently. [...] This would strengthen the findings on EDA mRNA reported in Fig 2i-j.

We agree that the role of the EDA-A2 ligand requires further expansion and emphasis within our manuscript. In response to the reviewer's suggestions, we have addressed this point in two ways:

- identifying cell types expressing EDA-A2 in blood
- determining whether EDA-A2 increases at protein level with aging or aging-associated conditions

These are, in fact, two different aspects. The total protein amount of EDA-A2 ligand measured in plasma may originate from various secretory organs such as the liver, thyroid, or adrenal glands. In contrast, the observed increase in EDA-A2 mRNA levels in blood must originate from blood cell populations. Hence, we quantified isoform-level gene expression of EDA-A2 by reprocessing the Monaco dataset (10.1016/j.celrep.2019.01.041, GSE107011), a reference for annotating blood cells across 29 cell types. Details regarding the analysis pipeline are now provided in the supplementary materials. Our analysis revealed that the transcription of EDA-A2 isoform is higher in T-cells, particularly T-helper and Naïve CD4/CD8 T-cells, indicating that these populations mostly contribute to the EDA-A2 mRNA expression levels observed in blood. Because EDA-A1 and EDA-A2 transcripts are almost identical and differ only for six nucleotides, only those reads spanning this specific six-nucleotide stretch can be used to distinguish between the two isoforms unambiguously.

Supplementary Table 13: indicated are isoform-level expression values (TPM) of EDA-A2 transcript (ENST00000374553) across 29 different blood cell types as determined from the dataset from Monaco et al.

ID	cell type	LIN	EDA-A2 (TPM)
Exhausted B cells	B cells	lymphoid	0.00
Naive B cells	B cells	lymphoid	0.00
Non-switched memory B cells	B cells	lymphoid	0.00
Switched memory B cells	B cells	lymphoid	0.00
Natural killer cells	NK cells	lymphoid	0.00
Plasmablasts	PB	lymphoid	0.00
Plasmacytoid dendritic cells	pDCs	myeloid	0.00
Central memory CD8 T cell	T cells	lymphoid	0.00
Effector memory CD8 T cells	T cells	lymphoid	0.00
Follicular helper T cells	T cells	lymphoid	0.00
MAIT cells	T cells	lymphoid	0.00
Non-Vd2 gd T cells	T cells	lymphoid	0.00
Terminal effector CD4 T cells	T cells	lymphoid	0.00
Terminal effector CD8 T cells	T cells	lymphoid	0.00
Naive CD4 T cells	T cells	lymphoid	0.88
Naive CD8 T cells	T cells	lymphoid	0.23
Th1 cells	T cells	lymphoid	0.42
Th1/Th17 cells	T cells	lymphoid	0.80
Th17 cells	T cells	lymphoid	0.09
Th2 cells	T cells	lymphoid	0.39
T regulatory cells	T cells	lymphoid	0.35
Vd2 gd T cells	T cells	lymphoid	0.00
Basophils	Myeloid	myeloid	0.00
Classical monocytes	Myeloid	myeloid	0.00
Intermediate monocytes	Myeloid	myeloid	0.00
Non classical monocytes	Myeloid	myeloid	0.17
Myeloid dendritic cells	Myeloid	myeloid	0.00
Neutrophils	Myeloid	myeloid	0.00
Progenitor cells	HSC	precursor	0.00

precursor
myeloid
lymphoid

Next, we aimed to determine if the transcriptional increase of the EDA-A2 isoform in blood extends to the protein level. Since the two isoforms of EDA (EDA-A1 and EDA-A2) differ by only two amino acids, mass spectrometry does not distinguish between them. Therefore, high-throughput protein-level analyses may encompass both isoforms. Given this limitation, we examined the dataset from Lehallier et al. and found that total EDA levels in plasma remain stable with increasing age (**Supp.Fig7**), suggesting that blood-produced EDA may represent a small fraction of the circulating EDA.

SUPPLEMENTARY FIGURE 7

Correlation between age and EDA protein in males and females
Scatterplots showing correlation between EDA abundance and age. Protein abundance was determined in plasma samples from 5,676 adults as described in (10.1038/s41586-023-06802-1) (https://twc-stanford.shinyapps.io/aqing_plasma_proteome_v2).

It is also possible that more sensitive proteomic quantification technologies, such as protein sequencing, that can discriminate EDA-A1 from EDA-A2, are needed to reliably detect isoform specific changes. Nonetheless, steady levels of EDA in plasma, combined with the concurrent increase in its receptor EDA2R, support the potential for heightened activation of EDA2R/EDA-A2 signaling with age.

These observations prompted us to investigate whether aging-associated comorbidities, rather than aging itself, could influence the total amount of EDA-A2 levels in blood. While plasmatic levels of EDA remain stable in healthy aging, isoform-specific EDA-A2 protein levels were measured by ELISA Assays and spotted as being significantly increased in conditions such as insulin resistance, type 2 diabetes (T2D) and nonalcoholic fatty liver disease (j.cca.2019.09.009). These comorbidities are directly associated with increased risk of developing sarcopenic phenotypes (10.1186/s13098-018-0326-5, 10.3389/fendo.2021.782391, 10.1007/s41999-019-00216-x, 10.2147/DMSO.S410834, 10.1038/s41598-024-53112-1) and, intriguingly, sarcopenia itself was associated with heightened levels of EDA-A2 in mice (10.1128/MCB.24.4.1608-1613.2004).

Fig.2n) Health conditions and aging-associated comorbidities linked to increased levels of plasmatic EDA.

These observations suggests that aging-related conditions may exacerbate EDA2R pathway activation by increasing ligand levels, thereby adding a new dimension to our understanding of EDA2R/EDA-A2 signaling in the context of aging. In fact, we find this information to be particularly relevant for the discussion of our paper, as it has a strong translational potential. We thank the reviewer for putting us in the direction of discussing this within our manuscript.

It is not clear if these results represent EDA2R protein expression in multiple tissues or just plasma samples. The results in this figure can be better described. If the data points correspond to different tissues, then a separate graph for each can be provided. Would it be possible to do a similar analysis for EDA/EDA-A2?

We acknowledge that the legend of this supplementary figure may have been confusing. Indeed, the EDA2R protein quantification was performed exclusively in plasma samples from 5,676 adults, including both males and females. We revised the figure legend to make this clear.

The authors demonstrated that EDA2R was upregulated in the aortic artery of the HGPS mouse model. The significance of this observation made in the aortic artery should be discussed in the main text.

We acknowledge that this is indeed an interesting point. The upregulation of *EDA2R* in the aortic artery of HGPS mice is relevant for several reasons, given that the aorta is the major blood vessel and plays a crucial role in cardiovascular health. Indeed, aging and age-related diseases are often associated with vascular dysfunction, including changes in the structure and function of the aortic artery. The increased expression of *EDA2R* may reflect an underlying mechanism by which aging and progeroid conditions contribute to vascular pathology, given that *EDA2R* signaling is implicated in inflammatory processes and apoptosis, both of which are critical in the development of atherosclerosis and other vascular diseases. Therefore, the increased expression of *EDA2R* in the aortic artery suggests a potential link between *EDA2R* signaling and vascular aging. We now discussed this interesting aspect in the main text.

Would it be possible to test *EDA2R* expression in other tissues of HGPS mice using additional datasets?

There were indeed a limited number of RNA-Seq datasets available involving progeroid mice at the time of our original submission. Therefore, to address the reviewer's request regarding the expression of *EDA2R* in other tissues of HGPS mice, we have extended our search to also include microarray experiments. Hence, in addition to the already analyzed sequencing experiment, we could identify 2 additional datasets, namely GSE67288 and GSE51202 (**Fig.1e**).

- GSE67288 (tissue: *keratinocytes*, organism: *Mus Musculus*, technology: *microarray*)
In this dataset we could retrieve expression profiles of skin keratinocytes samples from 8 HGPS samples and 8 genotype negative littermates. Following gene-expression analysis we performed gene-expression analysis we could determine that *EDA2R* mRNA expression was strongly increased compared to wild-type counterpart.
- GSE51202 (tissue: *white adipose tissue*, organism: *Mus Musculus*, technology: *microarray*)
We could retrieve expression profiles of white adipose tissue (WAT) samples from 3 HGPS (G609G/G609G) mice, 3 wild-type (WT) mice, and 3 mice (LCS/LCS) with isoform specific transcription of lamin C (a non-progeroid splicing variant of *LMNA*). We confirm that *EDA2R* expression is higher in HGPS mice.

Figure 1e

Left, boxplots comparing expression levels of *Eda2r* between HPGS and age-matched wild-type mice in murine keratinocytes (GSE67288) and gonadal adipocytes (GSE51203) (right). P-values were calculated using the Wilcoxon Rank-Sum test.

Our findings indicate that *EDA2R* overexpression is indeed consistent across multiple tissues in progeroid HGPS mice. This widespread upregulation reinforces the notion that *EDA2R* is a key player in the aging process, affecting various tissues and cell-types.

4+5. The authors performed a high-resolution analysis of *EDA2R* expression in muscle tissue of mice using a dataset of a single-cell RNA-sequencing study and reported high *EDA2R* expression in satellite cells. However, how *EDA2R* expression in these cells changes upon aging was not reported. There is at least one single-cell RNA-sequencing study comparing young and old muscle samples: Zhang et al (PMID: 36147777). In fact, this paper reported (using bulk RNA-sequencing) increased *EDA2R* expression in muscle samples from old humans and old mice (Figure 6a and 7i). This work should be cited. While single-cell RNA sequencing is used to study gene expression in mono-nucleated cells of muscle tissue, it is not ideal for studying myonuclei of multinucleated muscle fibers, which are underrepresented due to cell isolation. Instead, single-nucleus RNA-sequencing analysis of muscle samples would better represent gene expression in muscle fibers. In fact, age-dependent *EDA2R* expression has already been reported in a study utilizing single-cell RNA-sequencing. Perez et al (PMID: 36516485) demonstrated that *EDA2R* transcript was induced significantly in muscle biopsies of elderly subjects compared with young controls and there was a trend for increased *EDA2R* expression in the sarcopenic subjects. Furthermore, this study also identified a population of *EDA2R*+ myonuclei only present in the fast muscle fibers of older adults. This work should be cited as well.

We thank the reviewer for directing us into these interesting research studies, both papers (Zhang et al; Perez et al) have now been cited as they are functional to the discussion of our manuscript.

Perez et al identified *EDA2R* among the genes commonly upregulated in aged human and rat muscle. Particularly intriguing is that the authors identified an *EDA2R*+ cell population (CDKN1A/MYH8/COL19A1/LRRK2/*EDA2R*+) consistent with senescence and which was present only in samples obtained from older adults. Zhang et al also found *EDA2R* to be upregulated in the aging muscle

by bulk RNA-Seq. Within the same paper, single-cell sequencing results indicated expression of *EDA2R* in multiple cell-types, including stellate cells, which backups the relevance of our experiments aimed at evaluating the effects of *EDA2R* overexpression in these myogenic precursors. However, an association at the single-cell resolution of *EDA2R* expression with aging was not reported. Indeed, when we looked at the associated data set (**Supp.Fig5b**), quantified expression of *EDA2R* was not sufficient to draw any conclusion, and greater sequencing was needed. Hence, we investigated single-cell sequencing data made available by a recent publication from Lai et al. (PMID: 38649488) which includes both single-cell and single-nuclei sequencing, thereby also solving the problem related to the underrepresentation of polynucleated cells. First, we confirmed once more that *EDA2R* expression is abundant in myogenic precursors, myotubes and other cell-types. Very importantly, we could clearly determine that *EDA2R* expression is increased with age in all cell-types considered. Indeed, this once more supports that the increase of *EDA2R* with age is a behavior that can be generalized to different cells of origin. We found this information particularly useful for the scope of our paper and embedded it in the main figures (**Fig.2e**)

Fig.2e UMAP showing single cells from human limb skeletal muscles, as quantified by Lai et al. (2024), including over 387,000 cells/nuclei from individuals aged 15 to 99 years. **Left:** Major cell types annotated (MF = Myofiber). **Center:** Expression levels of *EDA2R*. **Right:** z-scores indicating the relative increase of *EDA2R* expression across cell types between young and old individuals.

6. In Figure 2e, an *EDA2R* plasmid was introduced into C2C12 myoblasts. However, *EDA2R* overexpression is very mild (< 2 fold). Therefore, gene expression changes may not reflect the full potential of the effect of *EDA2R* on the myoblast transcriptome. The authors may consider viral gene delivery. A similar experiment may also be considered in differentiated myotubes.

To address the reviewer's comments, we proceeded in two directions:

- First, we extended our experiments and analyses in the human setting using myoblasts derived from human donors, given that in our hands, the human plasmid results in higher *EDA2R* expression. In this context, we repeated the experiments adopting two different techniques, transfection and electroporation. Experiments were performed in triplicates and subjected to RNA-Sequencing. Comparable effects are achieved by transfection or with electroporation. Indeed, in both scenarios we obtained impressive increase of *EDA2R* in a range between 10-12 fold (**Supp.Fig5i**). The results obtained using both techniques strongly paralleled those obtained in mice, which one side indicates that the observed transcriptional effects following *EDA2R* overexpression can be extended from mice to humans, and that the mild increase of *Eda2r* in mice is sufficient to trigger the activation of downstream signaling.

Supp.Fig5e Boxplots comparing gene-expression (counts per million) in control vs *EDA2R* overexpressing human myoblasts transfected either by lipofectamine (Fold-induction = 11.2; FDR = < 2.47e-324) or electroporation (Fold-induction = 10.6; FDR = < 2.47e-324). Adjusted p-values were computed using Wald statistics as implemented in DESeq2.

- Next, we aimed to determine whether the results obtained can be generalized from myoblast progenitors to differentiated myotubes. Therefore, experiments were performed in C2C12-derived murine myotubes (**Methods**) which were transfected with either *Eda2r* or *GFP* plasmids. Samples subjected to RNA-Seq showed that the transcriptional parainflammatory responses observed in the myogenic progenitors are indeed extensible to myotubes. We feel that this information is valuable and therefore we decided to integrate it into the manuscript (**Fig.5f-h**).

Fig2. - f) Age-dependent (column 1), *EDA2R/Eda2r* overexpression-dependent (columns 2-4) enrichments in Hallmark and Reactome inflammation-related gene sets, as computed in human myoblasts (column 2), C2C12 murine myoblasts (column 3), and C2C12 differentiated murine myotubes (column 4). **g)** Heatmap representing genes significantly upregulated ($FDR < 0.05$) in both murine myoblasts and differentiated murine myotubes following *Eda2r* overexpression. Highlighted genes exhibit higher levels of fold induction. **h)** Resulting protein-protein interaction network (String-DB). Genes whose induction is confirmed in human myoblasts following *EDA2R* overexpression are indicated in yellow.

Overall, experiments conducted in myoblasts and myotubes in both mice and humans, indicate that *Eda2r/EDA2R* overexpression results in the perturbation of the same signaling pathways, including induction of chemotaxis and inflammatory responses, as well as repression of pathways associated with muscle development and protein catabolism (**Supp.Fig5h, Fig.2i**).

Supp.Fig5h) Enrichment map depicting significant induction or repression of gene sets from the comparison of transcriptional profiles of *EDA2R*-overexpressing vs. wild-type murine C2C12-differentiated into myotubes. **Fig.2i)** Enrichment map depicting significant induction or repression of gene sets from the comparison of transcriptional profiles of *EDA2R*-overexpressing vs. wild-type human myoblasts. - Each dot of h) and c) represents an individual gene set, with the size of dots proportional to the size of each gene set. Edges between nodes represent shared genes between adjacent pathways. The analysis includes gene sets from the Hallmark, Reactome, Wikipathways, and GeneOntology (BP-biological processes) collections.

During the revision process, we had initially opted for viral delivery which however resulted in potent interferon responses triggering senescence both in *EDA2R* and control vectors, which strongly complicates the transcriptomic readout, given that we are focusing on inflammatory pathways. Transfection of control plasmids through electroporation did not result in inflammatory or target effects. Therefore, given the strong overexpression of *EDA2R* we decided to use this technique to confirm the result obtained using other transfecting agents (lipofectamine).

7. Any correlation between EDA2R expression and sarcopenia in the elderly was not discussed. It is hard to find datasets with phenotypic parameters (i.e. muscle mass or physical performance), however, such results would significantly enhance the novelty of the manuscript.

We acknowledge with the reviewer that this is an important point to discuss. Bilgic et al. (Nature, 2023 PMID: 37165186) determined that elevated EDA2R expression in skeletal muscle is crucial in promoting cancer-induced muscle loss, which underscores an association between increased *EDA2R* and muscle weakness and atrophy, which mirror key features of sarcopenia. Interestingly, Perez et al. identified a trend towards increased *EDA2R* in sarcopenic patients and thank the reviewer itself for directing us towards this publication, and we embedded this reference within our discussion. It is challenging however to find large age-matched cohorts for comparison that include both phenotypic parameters such as muscle mass or physical performance and *EDA2R* expression data. In this context, it would be particularly interesting to assess EDA2R abundance at the protein level. We recognize the importance of this approach and plan to pursue it in follow-up studies.

In addition, increasing EDA2R/EDA-A2 signaling can be sustained both by the increase of *EDA2R* but also by the sole increase of its ligand. Conversely to *EDA2R*, there is supportive data showing that increasing plasmatic levels of EDA leads to multifocal myodegeneration in mice (Newton et al. 2023, PMID: 14749376). This information is particularly important, given that augmented EDA is sufficient to activate downstream signaling as indicated by our experiments performed in human myoblasts supplemented with EDA-A2. Indeed, we observed that transcriptional perturbations in human myoblasts treated solely with EDA-A2 resembled those obtained when *EDA2R* is overexpressed (**Fig.2m**). We have included this perspective in the revised manuscript, acknowledging the need for future work to explore the possible increase of EDA2R in sarcopenic patients, particularly at the protein level.

8. Please identify explicitly the human dataset used in the gene set enrichment analysis of Figure 2g. The human dataset used to determine age-associated increase of inflammatory pathways is the GTEX dataset (v8). We specifically focused on samples derived from skeletal muscle (gastrocnemius). We added this information in the figure legend.

9. Please correct the typos in line 111 and 196. The text was modified accordingly.

Reviewer #3 (Remarks to the Author):

The study by Barbera et al identifies a strong association between age and increased transcript levels of EDA2R, the receptor for the EDA2 isoform of Ectodysplasin A. This is observed in virtually all tissues in human, mice and rats. EDA2R transcripts are also increased in a mouse model of accelerated ageing. The authors further study the association of EDA2R transcripts with age in muscles and blood. In muscles, the correlation is confirmed. In human blood, EDA2R is apparently not as strongly increased as in other tissues, but transcripts levels of EDA, especially those of the EDA2 splice variant, the ligand of EDA2R, increase with age.

Further mining of datasets indicate correlation of EDA2R transcript expression with CRP, an inflammatory marker, and the authors suggest that EDA2R might be linked to an age-related inflammatory status, that they experimentally support by overexpression of EDA2R in a cell line and by measuring induction of inflammation-related genes (e.g IL6, IL1a).

Authors seem to have followed rigorous protocols to maximize reliability in sample comparison. The observation that EDA2R is increased with age is amply supported by data and is very interesting, especially in the context of EDA2R being implicated in muscle loss. The correlation with an inflammatory status is also interesting, although the experimental "proof" that EDA2R contributes significantly to inflammation by overexpressing EDA2R in a cell line is rather weak: EDA2R is not the only inducer of NF- κ B. It is probable that several other members of the TNFR family (to which EDA2R belongs) would have induced similar genes and pathways upon overexpression, all of which have been described to respond to NF- κ B activation.

We agree with the reviewer that overexpression of other members of the TNFR superfamily could similarly lead to enhanced inflammatory signaling. However, the uniqueness of EDA2R lies in its consistent increase with age, regardless of tissue type or sex, across multiple species.

To underscore this distinctive behavior of EDA2R compared to other TNF receptors, we analyzed the expression patterns of all annotated Tumor Necrosis Factor receptors in humans (GTEx dataset). We included death-domain containing receptors (*TNFRSF1A*, *FAS*, *TNFRSF25*, *TNFRSF10A*, *TNFRSF10B*, *TNFRSF21*), TRAF-interacting motif containing receptors (*TNFRSF1B*, *CD40*, *TNFRSF8*, *CD27*, *LTBR*, *TNFRSF4*, *TNFRSF9*, *TNFRSF13C*, *TNFRSF17*, *TNFRSF13B*, *TNFRSF11A*, *TNFRSF16*, *TNFRSF14*, *TNFRSF18*, *TNFRSF19*, *EDAR*, *EDA2R*, *RELT*, *TNFRSF12A*), and tethering receptors (*TNFRSF10C*, *TNFRSF10D*, *TNFRSF6B*, *TNFRSF11B*).

The obtained results highlight *EDA2R* as a clear outlier in its correlation with aging. Specifically, *FAS*, the second-highest ranking TNF receptor based on its correlation with age, has a median correlation coefficient of 0.188, which is less than half of the 0.421 correlation coefficient observed for *EDA2R*. We feel that this information is particularly of interest, hence we created a new figure panel (**Supp.Fig5a**).

Supp.Fig5a) Violin plot depicting the distribution of median Pearson's correlation coefficients between age and mRNA expression across tissues (GTEx dataset). Indicated are *EDA2R* and all other annotated Tumor Necrosis Factor receptors in humans, including death-domain containing receptors, TRAF-interacting motif containing receptors, and tethering receptors.

I would like to raise two points.

1.- As mentioned in the manuscript, the ligand EDA exists in at least two splice variants: EDA2 that binds EDA2R, and EDA1 that binds EDAR. Given the age-related increase of EDA transcripts in blood (Fig 2e), it would be interesting to indicate transcript levels of EDA in other tissues also, to know whether upregulation of EDA of Fig 2e is a tissue-specific or a general effect. EDAR could also be shown as a control (which I assume will not appear in top transcripts associated with ageing). This should be done at least for Fig 1a.

We determined that transcription of *EDA* mRNA is significantly correlated with increasing age in blood (Fig2k). In this context, *EDA-A2* isoform is predominant compared to *EDA-A1* (Fig2l). Unlike *EDA2R*, the association between *EDA* gene expression and age is not widespread across tissues. This absence of correlation also extends to the other ectodysplasin-A receptor *EDAR*, which, as suggested by the reviewer, serves as an interesting control for comparison. We have integrated this information into the main figures, now highlighting correlations with age in individual tissues (Fig1a) and in pan-tissue averages (Fig1b).

Fig.1a) Boxplots representing Pearson's correlation coefficients between mRNA expression of all genes with age, from GTEx dataset. Tissues were evaluated separately. Relative positioning of *EDA2R*, *EDAR* and *EDA* are indicated. For rankings exceeding the 50th, the reference quartile is reported (Q1-Q4). **b)** Representation of the median Pearson's correlation coefficient for each gene (dots) in all solid tissues. Red and blue dots represent genes increasing or decreasing with age, respectively.

2.- The source of EDA transcripts in blood is not immediately obvious. It would be an interesting addition to the manuscript if the authors could analyse a data set comparing human blood cell types and pinpoint the potential source of EDA transcripts in blood. As this experiment will not change the conclusion that *EDA2R* increases with age, I do not insist to have it done, should it present difficulties.

We acknowledge the reviewer's point regarding the source of *EDA* transcripts in blood, as this aspect is of particular interest. Indeed, while the total protein amount of the *EDA-A1/A2* ligands quantified in plasma may originate from various secretory organs such as the liver, thyroid, or adrenal glands, the observed increase in *EDA-A2* mRNA reflects transcription by blood cell populations. To pinpoint the responsible cell types, we determined isoform-level gene expression of *EDA-A2* through the analysis of the Monaco et al dataset (GSE107011), which is a key reference for blood cell types frequently used for annotating cell populations in single-cell experiments. Using computational approaches specifically tailored to quantify transcriptional isoforms (please see updated **methods**), we could estimate the abundance of *EDA* splicing variants, and focused on the transcriptional isoforms **ENST00000374552**, which codes for the 391 amino

acid-long EDA-A1, and **ENST00000374553**, which codes for the 389 amino acid-long EDA-A2. The specificity of these splicing variants to code for EDA-A1 and EDA-A2 is confirmed in the amino acid sequence by the absence of Glu 308 and Val 309 in the ENST00000374553-derived protein.

Figure for the reviewer: Multiple sequence alignment of the protein sequences of EDA-A1 (391aa) and EDA-A2 (389aa) encoded respectively by the two transcriptional isoforms ENST00000374552 and ENST00000374553. Alignment was performed using clustal omega algorithm (<https://www.ebi.ac.uk/jdispatcher/msa/clustalo>). Highlighted are the differences between the 2 protein sequences (Glu 308 and Val 309) which are lacking in the isoform ENST00000374553 that code for EDA-A2.

Transcript ID	Name	Protein	Biotype	CCDS
ENST00000374552.9	EDA-203	391aa	Protein coding	CCDS14394
ENST00000374553.6	EDA-204	389aa	Protein coding	CCDS43966
ENST00000616899.1	EDA-213	259aa	Protein coding	
ENST00000524573.5	EDA-208	386aa	Protein coding	CCDS35319
ENST00000338901.4	EDA-201	147aa	Protein coding	
ENST00000527388.5	EDA-210	148aa	Protein coding	CCDS55436
ENST00000525810.5	EDA-209	135aa	Protein coding	CCDS35318
ENST00000503592.5	EDA-206	136aa	Protein coding	
ENST00000502251.5	EDA-205	No protein	Protein coding CDS not defined	
ENST00000510881.2	EDA-207	No protein	Protein coding CDS not defined	
ENST00000530321.1	EDA-211	No protein	Protein coding CDS not defined	
ENST00000533317.5	EDA-212	No protein	Retained intron	
ENST00000374548.5	EDA-202	No protein	Retained intron	

Information for the reviewer: Similarly to the aminoacidic sequences, EDA-A1 and EDA-A2 transcripts are almost identical, differing by only six nucleotides. Due to the relatively short length of sequencing reads, only those reads spanning this specific six-nucleotide stretch can be used to distinguish between the two isoforms unambiguously.

Our analysis revealed that EDA-A2 expression is higher in T-cells, particularly across T-helper and Naïve CD4/CD8 T-cells, suggesting that this population contributes to the age-dependent increase in EDA-A2 mRNA levels that we observed in blood. These findings have been included in **Supplementary Table 13** and discussed within the manuscript.

Supplementary Table 13: indicated are isoform-level expression values (TPM) of EDA-A2 transcript (ENST00000374553) across 29 different blood cell types as determined from the dataset from Monaco et al.

ID	cell type	LIN	EDA-A2 [TPM]	
Exhausted B cells	B cells		0.00	precursor
Naive B cells	B cells		0.00	myeloid
Non-switched memory B cells	B cells		0.00	lymphoid
Switched memory B cells	B cells		0.00	
Natural killer cells	NK cells		0.00	
Plasmablasts	PB		0.00	
Plasmacytoid dendritic cells	pDCs		0.00	
Central memory CD8 T cell	T cells		0.00	
Effector memory CD8 T cells	T cells		0.00	
Follicular helper T cells	T cells		0.00	
MAIT cells	T cells		0.00	
Non-Vd2 gd T cells	T cells		0.00	
Terminal effector CD4 T cells	T cells		0.00	
Terminal effector CD8 T cells	T cells		0.00	
Naive CD4 T cells	T cells		0.88	
Naive CD8 T cells	T cells		0.23	
Th1 cells	T cells		0.42	
Th1/Th17 cells	T cells		0.80	
Th17 cells	T cells		0.09	
Th2 cells	T cells		0.39	
Tregulatory cells	T cells		0.35	
Vd2 gd T cells	T cells		0.00	
Basophils	Myeloid		0.00	
Classical monocytes	Myeloid		0.00	
Intermediate monocytes	Myeloid		0.00	
Non classical monocytes	Myeloid		0.17	
Myeloid dendritic cells	Myeloid		0.00	
Neutrophils	Myeloid		0.00	
Progenitor cells	HSC		0.00	

To further confirm our results, we interrogated The Database of Immune Cell Expression (DICE), a valuable resource for determining blood cell population-specific gene expression. However, the raw data is not freely accessible, and the processed gene expression quantification does not allow discrimination of transcriptional isoforms. Nonetheless, we observed that transcription from the *EDA* locus is mostly associated with T-helper/CD4 T-cells, which aligns with our previous observations.

Figure for the reviewer: *EDA* expression values stratified by blood cell-types as determined from the Database of Immune Cell Expression (DICE). Expression values are indicated in Transcript per million (TPM).
source: (<https://dice-database.org>)

Next, we aimed to determine if the transcriptional increase of the *EDA-A2* isoform in blood is reflected at the protein level. Due to the fact that *EDA-A1* and *EDA-A2* isoforms differ by only two amino acids, mass spectrometry cannot differentiate between them. Consequently, high-throughput protein-level analyses may apply to both isoforms. Given this limitation, we examined the dataset from *Lehallier et al.* and found that total *EDA* levels in plasma remain stable with increasing age (**Supp. Fig. 7**), suggesting that blood-produced *EDA* may represent a small fraction of the circulating *EDA*.

SUPPLEMENTARY FIGURE 7

Correlation between age and *EDA* protein in males and females
Scatterplots showing correlation between *EDA* abundance and age. Protein abundance was determined in plasma samples from 5,676 adults as described in (10.1038/s41586-023-06802-1) (https://twc-stanford.shinyapps.io/aging_plasma_proteome_v2).

It is also possible that more sensitive proteomic quantification technologies, such as protein sequencing, that can discriminate *EDA-A1* from *EDA-A2*, are needed to reliably detect isoform specific changes. Despite this, the stable levels of total *EDA* in plasma, coupled with the concurrent increase in its receptor *EDA2R*, support the potential for enhanced activation of *EDA2R/EDA-A2* signaling with age.

These findings prompted us to further investigate whether aging-associated comorbidities, rather than aging itself, could influence the total amount of *EDA-A2* levels in blood. While plasmatic levels of *EDA* remain stable in healthy aging, isoform-specific *EDA-A2* protein levels were measured by ELISA Assays and spotted as being significantly increased in conditions such as insulin resistance, type 2 diabetes (T2D) and nonalcoholic fatty liver disease (j.cca.2019.09.009). These comorbidities are directly associated with increased risk of developing sarcopenic phenotypes (10.1186/s13098-018-0326-5, 10.3389/fendo.2021.782391, 10.1007/s41999-019-00216-x, 10.2147/DMSO.S410834, 10.1038/s41598-024-53112-1) and, intriguingly, sarcopenia itself was associated with heightened levels of *EDA-A2* in mice (10.1128/MCB.24.4.1608-1613.2004). These observations suggests that aging-related conditions may exacerbate *EDA2R* pathway activation by increasing ligand levels, thereby adding a new dimension to our understanding of *EDA2R/EDA-A2* signaling in the context of aging.

Fig.2n) Health conditions and aging-associated comorbidities linked to increased levels of plasmatic EDA.

In conclusion, we strongly expanded the discussion respective to EDA2R ligand given that we information to be particularly relevant for the discussion of our paper. We thank the reviewer for putting us in the direction of discussing this within our manuscript.

Minor points

a) Fig 1b. The authors have shown names of 6 age-associated transcripts in addition to EDA2R. They could also identify the most downregulated transcript.

We now updated the figure panel to also include labeling of the most downregulated genes. A comprehensive list of all genes, including p-values and signed correlation coefficients (for each tissue separately and averages) are provided in **Supplementary Table 1** and **Source data** file.

b) Methods, line 243. Please indicate the % of FBS used to culture C2C12 cells.

C2C12 (ATCC) were grown in DMEM supplemented with 10% fetal bovine serum (FBS) (Euroclone, Pero, Italy) and 2 mM L-glutamine (Gibco), and maintained in culture at 37 °C with 5% CO₂, without reaching confluence to avoid fusion into myotubes. We updated the methods accordingly.

c) The last sentence of the main text suggests that EDA2 - EDA2R could be targeted to counteract ageing. Whether this will work remains highly speculative. The cautious way in which this sentence is written is appropriate, but a stronger statement should not be made.

We edited the text accordingly, emphasizing however the current lack of established small molecules specifically acting as EDA2R antagonists.

Reviewer #4 (Remarks to the Author):

The study presents a comprehensive investigation into the role of Ectodysplasin-A2-Receptor (EDA2R) in ageing, identifying it as a tissue-independent biomarker associated with ageing across different species. Through extensive bioinformatics analysis of various datasets and experimental validation, the authors conclude that EDA2R expression increases with age and is linked to parainflammatory responses and sarcopenia-like phenotypes. Targeting EDA2R signaling is therefore proposed as a novel approach to mitigating age-related physiological changes.

Suggestions for Improvement or Clarification:

1. **Methodological Transparency:** while the study uses a rigorous multi-step bioinformatics approach, the statistical methods and criteria for data filtering (beyond FDR adjustment) could be better detailed to enhance reproducibility (i.e. expanding the information given in lines 261 to 298, maybe in supplementary).

To address the reviewer's query requesting more details about the reproducibility of the analyses conducted, we have restructured the methodology outlined in lines 261-298, and concurrently provided a more detailed description of the steps performed for the analysis in the methods section. **To enable complete reproducibility, we generated a source file (SourceData.xlsx) which is now deposited along with the manuscript, containing numeric values used for generating each figure panel.**

Furthermore, all source code used in the original submission and throughout the revision process is now deposited at our GitHub repository (<https://github.com/BolisLab/eda2r>).

We also restructured and added novel supplementary files which now contain following information:

- **Supplementary Table 1:** contains Pearson's correlation coefficients between gene-expression and chronological age of donor (human) across 26 different tissue types.
- **Supplementary Table 2:** contains p-values determined from n=1000 permutations, representing the likelihood of observing Pearson's correlation coefficients at least as extreme as the observed result reported in Supplementary File1.
- **Supplementary Table 3:** contains Pearson's correlation coefficients between gene-expression and chronological age of mice across 14 different tissues.
- **Supplementary Table 4:** contains Pearson's correlation coefficients between gene-expression and chronological age of rats across 11 different tissues.
- **Supplementary Table 5:** contains differential expression results of HGPS mouse models compared to aged-matched wild-type counterpart.
- **Supplementary Table 6:** contains quantification of *EDA2R* mRNA expression in human vastus lateralis muscle samples from dbGap (phs001048).
- **Supplementary Table 7:** contains differential expression results from microarray experiment (GSE52550), comparing the gastrocnemius muscle of aged mice to that of young mice.
- **Supplementary Table 8:** contains differential expression results of *Eda2r* mRNA expression changes in gastrocnemius muscle of aged rats compared to baseline (6 months)
- **Supplementary Table 9:** murine C2C12 myoblasts transfected with *Eda2r* vs matched controls (GFP)
- **Supplementary Table 10:** murine C2C12-derived myotubes overexpressing *Eda2r* vs matched controls (GFP)
- **Supplementary Table 11:** human primary myoblasts transfected with *EDA2R* vs matched controls (GFP)
- **Supplementary Table 12:** human primary myoblasts supplemented with EDA-A2 vs vehicle
- **Supplementary Table 13:** contains isoform-level expression of *EDA-A2* quantified in GSE107011 (Monaco et al), a reference for annotating blood cells across 29 cell types.

Additionally, we performed novel experiments during the revision process, specifically:

- We explored whether our results obtained in murine myoblasts could be extended from myogenic progenitors to terminally differentiated myotubes. RNA-Seq analysis demonstrated that the inflammatory responses observed in progenitors are confirmed in differentiated myotubes (**Figure 2f-h and Supplementary Figure 5f-h**).
- We have successfully translated our findings from mice to humans using primary myoblasts derived from human donors. These results were validated by transfection using both lipofectamine and electroporation, confirming the translational potential of our findings. Pathway analysis in this setting revealed that EDA2R overexpression is strongly associated with signaling pathways related to muscle development and myogenic differentiation (**Figure 2f,i Supplementary Figure 5i**).
- To further validate our findings without exogenous RNA manipulation, we treated human myoblasts with EDA-2 ligands. This treatment induced strong para-inflammatory responses and activation of downstream signaling, mirroring the effects of *EDA2R* overexpression, thereby providing further rationale for developing small molecules aimed at hindering the formation of the EDA2R/EDA-A2 complex (**Figure 2m**).
- We evaluated the effects of *EDA2R* knock-down by performing silencing experiments in human myoblasts. Our results revealed transcriptional perturbations leading to potent downregulation of pro-inflammatory cytokines and other biological pathways linked to parainflammatory responses (**Supplementary Figure 5j**).

All these experiments were subjected to RNA-Sequencing and made freely accessible at the European Nucleotide Archive (ENA):

- **E-MTAB-13101**: *Eda2r* overexpression in C2C12 murine myoblasts
- **E-MTAB-14234**: *Eda2r* overexpression in murine C2C12 cells differentiated to myotubes
- **E-MTAB-14239**: Experiments performed in human myoblasts involving *EDA2R* transfection by either lipofectamine or electroporation, treatment with the EDA-A2 ligand, and silencing of *EDA2R* through the action of short hairpins.

These comprehensive updates and additions significantly enhance the methodological transparency and reproducibility of our study, addressing the reviewer's concerns and providing a robust foundation for future research.

2. Mechanistic Insights: EDA2R's role in ageing is explored extensively, but the molecular mechanisms underlying EDA2R-induced parainflammatory responses could be further elucidated, potentially

through pathway analysis and interaction studies. In particular lines 128 to 130 could be expanded a little and supported with data or literature.

We agree with the reviewer that more mechanistic insights would be beneficial in the discussion of our paper. In particular, EDA-A2 has been shown to interact with the EDA2R receptor to activate the non-canonical NF- κ B signaling pathway (von Renesse et al 2023, Domaniku et al 2023), which is crucial for inflammation and immune responses. This activation involves the kinase NIK (NF- κ B-inducing kinase), leading to the cleavage of NF- κ B2 (p100) to its active form (p52), which forms transcriptionally active p52/RelB complexes. These complexes translocate to the nucleus to induce the expression of genes involved in inflammatory responses, including pro-inflammatory cytokines such as IL-6 and CXCL5. Conversely, the canonical NF- κ B pathway involves the activation of I κ B kinase (IKK) complex, which phosphorylates I κ B proteins, leading to their degradation and the release of NF- κ B dimers (usually p50/RelA) that translocate to the nucleus to initiate the transcription of target genes involved in immune and inflammatory responses. Thus, while both pathways activate NF- κ B and regulate inflammation, they do so through distinct mechanisms and result in the formation of different active NF- κ B complexes: p50/RelA(p65) in the canonical pathway and p52/RelB in the non-canonical pathway.

Therefore, given that EDA2R signaling has been previously associated to the activation of NF- κ B, we investigate whether the transcriptional responses observed in our data are induced by the activation of the canonical or non-canonical NF- κ B pathway. Hence, we specifically evaluated the transcription levels of the RelA/p50 and RelB/p52 complexes. These complexes serve as distinct transcriptional indicators for the activation state of their respective pathways, in contrast to downstream targets and cytokines, which may overlap between the two pathways. Our data clearly show that EDA2R overexpression in human and murine myoblasts, as well as in differentiated myotubes, results in a strong transcriptional increase in the RelB/p52 complex-associated genes (*RELB,NFKB2*), but not in the RelA/p50 complex-associated ones (*RELA,NFKB1*).

Supp.Fig5k Enrichment analysis (left) and fold-change increase (right) of key elements in the canonical (NFKB1+RELA) and non-canonical (NFKB2+RELB) NF- κ B signaling pathways following overexpression of EDA2R. Enrichments results are presented for human myoblasts (column 1), C2C12 murine myoblasts (column 2), and C2C12-differentiated murine myotubes (column 3).

This finding supports the literature that indicates the involvement of the non-canonical NF- κ B signaling pathway consequent to EDA2R-EDA/A2 activation. We incorporated this information in **Supp.Fig5k** and extended the discussion in our manuscript to include these findings and relevant literature.

Moreover, experiments we conducted in myoblasts and myotubes in both mice and humans, indicate that *Eda2r/EDA2R* overexpression results in the perturbation of the same signaling pathways, including chemotaxis and inflammatory responses, and notably, NK- κ B signaling (**Fig.2i**).

Fig.2i) Enrichment map depicting significant induction or repression of gene sets from the comparison of transcriptional profiles of *EDA2R*-overexpressing vs. wild-type human myoblasts. - Each dot of h) and c) represents an individual gene set, with the size of dots proportional to the size of each gene set. Edges between nodes represent shared genes between adjacent pathways. The analysis includes gene sets from the Hallmark, Reactome, Wikipathways, and GeneOntology (BP-biological processes) collections.

Finally, we now observed that triggering of para-inflammatory responses following overexpression of *EDA2R/Eda2r* is observed both in myogenic precursors and differentiated myotubes and this phenomenon extends from mice to humans.

Fig2. - f) Age-dependent (column 1), *EDA2R/Eda2r* overexpression-dependent (columns 2-4) enrichments in Hallmark and Reactome inflammation-related gene sets, as computed in human myoblasts (column 2), C2C12 murine myoblasts (column 3), and C2C12 differentiated murine myotubes (column 4). **g)** Heatmap representing genes significantly upregulated (FDR < 0.05) in both murine myoblasts and differentiated murine myotubes following *Eda2r* overexpression. Highlighted genes exhibit higher levels of fold induction. **h)** Resulting protein-protein interaction network (String-DB). Genes whose induction is confirmed in human myoblasts following *EDA2R* overexpression are indicated in yellow.

3. Therapeutic Implications: the potential for targeting EDA2R as a therapeutic strategy is mentioned and features and one of the most important conclusions of this work; however, while the authors provide clear support for the effects of overexpressing EDA2R, the study could benefit from either preliminary data on pharmacological inhibition or other interventions to lower EDA2R expression in ageing models to substantiate these claims.

Currently, there is no established antagonist specifically targeting EDA2R, which limits direct pharmacological interventions. However, we recognize the importance of exploring methods to modulate EDA2R activity. In our study, we performed experiments to silence EDA2R in human myoblasts.

Our results indicated that the reduction of EDA2R expression effectively decreased the activity of inflammatory signaling pathways and reversed the transcriptional perturbations induced by EDA2R overexpression in the same cellular model. While the development of small molecules specifically targeting EDA2R remains an area of interest, our findings contribute to the understanding of EDA2R's role and underscore its potential as a therapeutic target. The observed effects of EDA2R knock-down on inflammatory signaling highlight the therapeutic promise of targeting this receptor.

We are confident that our manuscript will stimulate further research and interest in developing EDA2R-specific inhibitors, which could provide new therapeutic avenues for managing inflammation and related conditions associated with aging. We agree that further studies, including pharmacological inhibition, would be valuable. Future work could focus on developing and testing potential EDA2R antagonists to complement our findings and substantiate the therapeutic implications of targeting EDA2R in aging models.

Supp.Fig5j) Enriched inflammation-related gene sets from the Hallmarks and Reactome collections, as determined in human myoblasts following overexpression (column 1) or knockdown (columns 2) of EDA2R, increasing or decreasing with age, respectively.

Minor Typos or Elements to address:

4. Consistency in Gene/Protein Naming: Ensure consistency in the use of gene and protein names (e.g., EDA2R vs. Eda2r) throughout the manuscript for clarity, making sure the two are indeed used when referring to gene (transcription, expression) or protein (protein levels, signalling).

We modified the manuscript to use uppercase when referring to human genes and lowercase with an initial capital letter when referring to mouse genes. Italics were used when discussing genes or transcription, while non-italic text was used for protein levels and signaling pathways.

5. Statistical Details: Provide more detailed statistical analysis information, including exact p-values and n numbers, especially in figure legends and the results description.

We edited the manuscript to include statistical details in all figure legends and made sure to always include sample size and exact p-values.

REBUTTAL (2nd revision) – POINT BY POINT REPLY TO REVIEWERS

Reviewer #2 (Remarks to the Author):

The authors have satisfactorily addressed all of my comments. This is very much appreciated. I only recommend the following correction regarding the legends of the supplementary figures 6 and 7.

Regarding my previous comment #2 on the supplementary figure 5 (now supplementary figure 6), the authors indicated that they revised the figure legend. The legend provided in the rebuttal is in fact corrected. However, the legend in the main text remains the same as the previous version. It reads as measurements were taken from multiple tissues while it should have been blood plasma only. This correction also applies to the new supplementary figure 7, where EDA expression was measured.

We apologize for the oversight in not updating the figure legends in the main text as intended. The legends for both Supplementary Figure 6 (**now Supplementary Figure 7**) and Supplementary Figure 7 (**now Supplementary Figure 8**) have now been accurately revised to reflect that the measurements were taken exclusively from blood plasma samples. We are grateful to the reviewer for acknowledging the revisions we have made and would like to express our thanks for the valuable insights provided throughout the revision process.

Reviewer #3 (Remarks to the Author):

The authors have adequately and thoroughly taken into consideration my comments/suggestions, among others by showing expression levels of EDA and EDAR in comparison to XEDAR, by investigating the possible source of EDA transcripts in blood, and by improving parts of the method section. Thank you.

We sincerely appreciated the reviewer's helpful comments and are pleased that our revisions have addressed the suggestions. The feedback received has been relevant in improving the clarity and depth of our study, and we are grateful for the reviewer's careful evaluation throughout this process.

Reviewer #5 (Remarks to the Author):

This is an interesting report and the authors did a good job at leveraging data in the public domain with experimental data that they generated for this study. The original comments of reviewer 4 were well addressed. However, I found some remaining issues with the analysis that could be addressed.

Fig1a: Showing the correlation between age and gene expression is not sufficient because the results could be biased by outliers or funny patterns, so supplement figure 2 is really critical. This figure actually shows that the claim of an increase with age pattern across all tissues is not precise. The scatter plots for brain, breast, liver, pituitary gland for example show that some of the correlation values may be driven by outliers. Did the authors remove outliers/bad samples before conducting the analysis?

We thank the reviewer for highlighting this important point regarding the potential influence of outliers in our analysis. We fully acknowledge the need for a more rigorous approach in addressing this concern, and to that end, we setup and employed a comprehensive multi-step process to ensure that the correlations between gene expression and age were not biased by outliers or irregular patterns.

Our approach is outlined below:

1. **Repeated Resampling:**

- Rather than calculating correlations using the entire dataset at once, we employed a leave-half-out (LHO) approach for each tissue. In this method, we randomly split the data into 10 subsets, each containing 50% of the original samples. This allowed us to estimate correlations separately for each of the 10 LHO subsets within each tissue. This

strategy minimizes the likelihood that outliers drive the overall results, as the random resampling ensures that outliers, if present, are not consistently selected in every subset.

2. **Outlier Removal:**

- For each of the 10 subsets in every tissue, we applied an additional outlier filtering step. Specifically, for each gene, we removed samples whose expression values deviated more than three standard deviations from the mean of that gene. This ensured that even if an outlier was randomly included in a particular LHO subset, it would not influence the estimated correlations.

3. **Correlation Estimation:**

- We used non-parametric Spearman's rank correlation to estimate the relationship between gene expression and age for each gene, across the different LHO subsets. The correlation estimates from the 10 subsets were then averaged to provide an aggregated correlation coefficient, which is now robust against the potential influence of outliers.

4. **Aggregation:**

- Once the resampling and correlation estimation steps were complete, we average the results across all tissues to produce a multi-tissue aggregate. This provides a comprehensive measure of the correlation between gene expression and age, ensuring that the findings are robust and unaffected by outliers or sampling artifacts.

To further clarify our workflow for the readers, we have generated a new figure panel, now included as **Supplementary Figure 1**. This figure visually represents the repeated resampling and outlier removal steps we implemented. This comprehensive pipeline, which integrates outlier detection and repeated resampling, provides a solid foundation for the observed correlations between age and gene expression.

Supp. Fig1) Illustrated is the multi-step process used to ensure robust correlation results between gene expression and age across tissues. The workflow includes: 1) Data Pre-processing and batch effect removal 2) Leave-Half-Out (LHO) resampling of the data into 10 subsets per tissue, each containing 50% of the samples, to reduce the impact of outliers. Next, remaining outliers are discarded, where samples exhibiting expression values deviating more than three standard deviations from the mean for each given gene are excluded; 3) Spearman's rank correlations are computed for each gene across the subsets, and the results are averaged for each tissue. Subsequently, mean correlations across tissues are calculated to provide a comprehensive result. This approach minimizes bias from outliers and ensures reliable correlation estimates.

Repeated 10 fold Leave Half Out resampling procedure:

1. Arrange the tissue-specific samples in a random order
2. For $i = 1, \dots, k=10$:
 - Select half of the samples randomly and discard others
 - for each gene: discard outliers > 3 SD from mean
 - Estimate Spearman's coefficient for each gene
3. Average coefficients and p-values for different k
4. Estimate multi-tissue average

Consequently, **Figure 1a** and **Figure 1b** have been appropriately updated to reflect the new analysis workflow. The correlation estimates obtained using this revised workflow are now provided in **Supplementary File 1**. For comparison, the original Pearson correlation coefficients are also included within the same table/file. Methods were also updated accordingly.

We thank the reviewer for its suggestion, which we believed was functional to further strengthen the reliability of our findings.

Related to this figure, the sentence between lines 487-489 needs clarification. What is the "biological age" they refer to?

The term 'biological age' in the original text was inadvertently misleading, as we were actually referring to the chronological age of the individual (i.e., the age in years since birth). To avoid confusion, we have corrected this in the figure legend, now referring specifically to 'chronological' age, which was used as a covariate in the design matrix.

Fig1b: why using the median correlation across tissues and not the mean?

In our revised analysis, as described in the updated workflow, we now use the mean correlation rather than the median. While the results were similar when using either approach, we agree that the mean provides a more appropriate representation in this context. Using the median may inadvertently downplay tissues where the correlation is less strong, whereas the mean offers a more complete aggregate view, capturing the full spectrum of tissue-specific correlations.

Fig 1: The legend of figure 1 should mention sample sizes used for the analysis.

We agree that this detail is important for transparency and context. In response, we have now included the sample sizes for each individual tissue directly within **Figure 1**, ensuring that this information is clearly communicated.

Fig2a: the sample size in the 20-40 age group is very small, so the robustness of the results is questionable. In addition, the data show clear outliers that may drive the significance levels. I would suggest conducting an analysis in which outliers are removed and maybe the age groups 20-40 and 40-50 merged into one.

We agree with the reviewer's observation regarding the limited sample size in the 20-40 age group, despite the results remaining statistically significant. Following the reviewer's suggestion, we generated a new version of the analysis (now provided as **Supplementary Figure 6b**), which aggregates the 20-50 age groups and removes outliers. Outliers were defined as samples with expression values

more than three standard deviations above the mean, in line with the outlier detection method used for our tissue correlation workflow.

Notably, significance was maintained and even improved in this updated analysis. We have opted to present this refined version without outliers in the supplementary material to avoid giving the impression to the readers that we excluded some samples just to enhance the statistical significance of our results. We now reference the **Supplementary Figure 6b** in the main text, and we acknowledge the underrepresentation of the 20-40 age group in the respective figure legend.

Fig 2d: I could not find the details of the scRNAseq data. The authors should provide some details of the analysis to show UMAP plots.

The data used for single-cell RNA sequencing was retrieved from publicly available sources, specifically from the Dryad repository (doi:10.5061/dryad.t4b8gtj34, McKellar et al. 2021) and the dataset from Zhang et al. (https://mayoxz.shinyapps.io/Muscle). These datasets were already processed, and we relied on the precomputed UMAP representations provided in the original publications, both of which were generated using principal component analysis (PCA) for dimensionality reduction. For the McKellar et al. dataset, a Harmony batch-corrected UMAP was provided, as described in their publication. Harmony was chosen for its robust ability to correct batch effects across diverse datasets—covering different ages, injury models, and experimental conditions—while preserving biological variability. For the dataset retrieved from Zhang et al., a standard UMAP without additional batch correction was used. We have now included these details in the methods section for clarity. Additionally, we have updated our GitHub repository to include the code necessary to replicate our figures, starting from the publicly available data, ensuring transparency and reproducibility.

Fig 2j: In addition to p-values, it would be useful to show effects in a standardized scale, so that they are comparable.

We agree that including effect sizes in Fig 2j would be beneficial. Indeed, we acknowledge that the original version is somewhat redundant, as both the dot size and color represented statistical significance. In the updated version, we have revised the figure by using dot size to reflect effect size (based on standardized coefficients) while retaining the color coding (violet for significant and grey for non-significant results). Both the p-values and coefficients are now provided in the associated source data file.

REBUTTAL (3rd revision) – POINT BY POINT REPLY TO REVIEWERS

Reviewer #5 (Remarks to the Author):

The authors have revised the manuscript to address all the issues that I raised and the revised manuscript is acceptable.

We sincerely thank the reviewer for their positive evaluation. We greatly appreciate the time and effort dedicated to reviewing our work, particularly in taking over from a previous reviewer. We are pleased that the revised manuscript meets the required standards.